

# TephAta - An online data collection of tephra data from the Atacama Desert

Niklas Leicher[1], Vincent Feldmar[2], Andrés Quezada Jara[3], Paulina Vásquez Illanes[3], Fernando Sepúlveda Vásquez[3], Frank Wombacher[1], Markus Lagos[4], Tanja Kramm[2], Gabriel González[5], Klaudia Kuiper[6], Christoph Breitkreuz[7], Alberto Sáez[8], Domingo Gimeno[9], Lluís Cabrera[8], Inés Rodríguez Araneda [10], Bernd Wagner[1], Georg Bareth[2], Volker Wennrich [1]

[1]Institute of Geology and Mineralogy, University of Cologne, Cologne, 50672, Germany
[2]Institute of Geography, University of Cologne, Cologne, 50672, Germany
[3]Servicio Nacional de Geología y Minería (SERNAGEOMIN), Avenida Santa María 0104, Providencia, Santiago, Chile
[4]Institute of Geosciences and Meteorology, University of Bonn, Bonn, 53115, Germany
[5]National Research Center for Integrated Natural Disaster Management, Departamento de Ciencias Geológicas, Universidad Católica del Norte, Antofagasta, Chile
[6]Department of Earth Sciences, Vrije Universiteit Amsterdam, Amsterdam, Netherlands
[7]Institut für Geologie und Paläontologie, Bernhard-von-Cotta-Str. 2, TU Bergakademie Freiberg, 09599 Freiberg, Germany
[8]Geomodels Research Institute, Department Dinàmica de la Terra i de l'Oceà, Universitat de Barcelona, Marti Franques s/n, 08028 Barcelona, Spain
[9]Department de Mineralogia, Petrologia i Geologia Aplicada, Universitat de Barcelona, 08028 Barcelona, Spain
[10]Departamento de Obras Civiles y Geología, Universidad Católica de Temuco, Temuco, Chile

*Correspondence to*: Niklas Leicher (n.leicher@uni-koeln.de)





**Abstract**

Tephrostratigraphy and -chronology are powerful tools using volcanic ash (tephra) layers for establishing stratigraphic correlations and/or to obtain chronological information for different kinds of sedimentary archives. To develop
tephrostratigraphic frameworks, which ideally document a most complete spatial and temporal record of regional volcanic activity, precise geochemical and chronological characterization of tephra layers is needed. Based on these frameworks, newly discovered tephra layers can be linked to retrieve stratigraphic and chronological information. Tephrostratigraphic frameworks exist or are under construction for various regions. However, for some regions of the world the potential of these methods is not exploited yet, although being influenced by long-term and frequent volcanism and tephra deposition. One of these regions
is the Atacama Desert of northern Chile. There geochemical compositions of Pleistocene tephra layers (volcanic glass) were recently systematically investigated for their stratigraphy and chronology within the scope of the Collaborative Research Centre 'Earth – Evolution at the Dry Limit' (CRC 1211). These analyses were accompanied by the development of a tephra database called TephAta, which aims at providing a long-term basis for tephrostratigraphic and -chronological data of the Atacama Desert. TephAta allows digital documentation of the full variety of tephra-related datasets within one database and
provides search functions to foster and simplify the continuous expansion of the regional framework following the FAIR (findability, accessibility, interoperability, and reusability) data principles. A first data compilation (Leicher N., 2027) of 106 tephra samples origin from 91 tephra deposits is now available and will be continuously extended spatially and temporally. TephAta does not only provide a stratigraphic and dating tool for paleoenvironmental or mapping studies, but it also helps to explore the integrity of the explosive volcanic history of the region along with the associated volcanic hazard and risk
assessment.

**1 Introduction**

Volcanic ashes (so-called tephra) are intercalated in many kinds of sedimentary successions and serve as ideal chronological and stratigraphic marker horizons. This fosters their use as targets for stratigraphic and chronological approaches, e.g., in geoscientific and archaeological studies (Lowe, 2011), by tephrostratigraphy and –chronology. The geochemical composition
of volcanic glass is the most common and reliable parameter (Lowe et al., 2017) to construct tephrostratigraphic frameworks by correlating tephra layers, and to identify volcanic sources. Due to aeolian fractionation during volcanic dispersal, volcanic glass often remains as the major eruptive component, and its relative abundance increases with increasing distance to the eruptive center. Due to the quenched character of glass, its geochemical composition closely resamples the composition of the magma at the time of eruption and thus can be used as a characteristic fingerprint.

Tephrostratigraphy and -chronology are well-developed and frequently used techniques in many regions of the world including the Mediterranean (e.g., Sulpizio et al., 2003; Wulf et al., 2004; Giaccio et al., 2019; Leicher et al., 2021; Vakhrameeva et al., 2021), Northern Europe and the N-Atlantic (e.g., Griggs et al., 2014; Lowe et al., 2015; Abbott et al., 2018), Central America (e.g., Kutterolf et al., 2016; Schindlbeck et al., 2018), New Zealand (e.g., Gehrels et al., 2006; Lowe et al., 2013; Hopkins et al., 2021) and eastern Asia (e.g., Sagawa et al., 2018; Albert et al., 2019; Portnyagin et al., 2020; Feng et al., 2022).
Tephrostratigraphic databases exist for specific regions, e.g., for (northern) Europe (TephraBase, Newton et al., 2007; Riede et al., 2011; RESET, Bronk Ramsey et al., 2015), the east African Rift (EarthD, Mana and Dimaggio, 2023), Kamchatka (TephraKam, Portnyagin et al., 2020), New Zealand (TephraNZ, Hopkins et al., 2021) southern Chile (BOOM!, Martínez Fontaine et al., 2023) or Antarctica (AnT, Kurbatov et al., 2014).

In some regions, however, where explosive volcanism has produced frequent and widely distributed tephra layers,
tephrostratigraphy and -chronology is not yet well established. Among these is the western side of the Andes in Central South America, including the Atacama Desert. The Central Volcanic Zone of the Andes (14°-28° S; De Silva, 1989b) has a long



volcanic activity (c. 240 Ma; Oliveros et al., 2018) and is known for major explosive eruptive events during the Neogene (c. 25-1 Ma, cf. Wörner et al., 2000; Burns et al., 2015; Van Zalinge et al., 2016). Moreover, the geodynamic evolution of the Andes, and its spatial and temporal differences in the thickness and composition of the crust, led to varying degrees of magma-

crust interactions and thus good preconditions for the magmatic differentiation of individual eruptions (Kay et al., 2010; Brandmeier and Wörner, 2016; Burns and De Silva, 2023). This is a prerequisite for determining eruption-specific geochemical fingerprints that allow inter-site correlations of tephra layers and the construction of tephrostratigraphic frameworks.

Despite dominating easterly wind systems, volcanic ashes can be found abundantly from the Andes to the Pacific coast (Marquardt et al., 2005; Vásquez and Sepúlveda, 2013; e.g., Breitkreuz et al., 2014). The long-term aridity of the Atacama

region, which is discussed to have started as early as ca. 25 Ma ago (Dunai et al., 2005; Evenstar et al., 2017), and the associated low degree of erosion (e.g., Ritter et al., 2023), ensured a good preservation of volcanic edifices and pyroclastic deposits in sedimentary archives. Nonetheless, tephra layers of the Atacama region hitherto were mainly directly dated to constrain paleoenvironmental and tectonic processes (Sáez et al., 2012; Kirk-Lawlor et al., 2013; Jordan et al., 2014) or during the extended geological mapping of the Chilean National Geology and Mining Survey (SERNAGEOMIN; e.g., Medina et al.,

2012; Blanco and Tomlinson, 2013; Sepúlveda et al., 2014; Vásquez et al., 2018). Only very few tephra layers were investigated for their glass geochemical fingerprint to establish local stratigraphic correlations (Placzek et al., 2009; Breitkreuz et al., 2014; Tapia et al., 2015). In contrast, rather intensive research has been carried out on the magmatic evolution on the volcanoes of the Andes (e.g., De Silva, 1989a; Wörner et al., 2000; Kay et al., 2010; Mamani et al., 2010). Comprehensive results of this research are made available within the Central Andes Geochemical and Geochronology database

(https://andes.gzg.geo.uni-goettingen.de/). This database provides basic information concerning the volcanic sources of tephra layers, but the type of data (whole-rock analyses) is less suitable to establish reliable correlations between tephra layers due to lateral variations in their crystal content (Tomlinson et al., 2012a). A tephrostratigraphic framework including the identification and detailed geochemical characterization of widespread marker horizons, which can be used for stratigraphic correlations and the transfer of ages between equivalent tephra layers, however, is currently missing at the western side of the Andes.

The Collaborative Research Centre "Earth – Evolution at the Dry Limit" (CRC 1211) aims at disentangling how the shaping of land-surfaces by past episodes of wetter climate coevolved with the evolution of life in arid environments (Dunai et al., 2020). The main working area of the CRC is the Atacama Desert in Chile (Fig. 1a). For its multidisciplinary endeavor, chronological control on the sedimentary climate archives and determination of dates and rates of surface processes are of fundamental importance. Furthermore, the scarcity of dateable material in the extreme desert environment of the Atacama

Desert limits the chronological information provided by other dating methods. One promising dating target are tephra layers, which are found in various sedimentary archives in the desert, including alluvial and fluvial deposits, lacustrine, salars and playa deposits (Sáez et al., 2012; Jordan et al., 2014; Vásquez et al., 2018; Medialdea et al., 2020; Ritter et al., 2022; Wennrich et al., 2025). Within the CRC1211, therefore, volcanic glass shards of tephra layers in the Atacama Desert were for the first time systematically investigated geochemically for tephrostratigraphic and tephrochronological purposes. The obtained data

is stored in the newly developed regionally focused tephra database TephAta. TephAta is designed to combine the vast variety of datasets that can be obtained for tephra samples ranging from field and lab metadata to morphological, geochemical, stratigraphic, and chronological datasets. Storing the data in the dedicated TephAta database will facilitate to focus on the development of a regional tephrostratigraphic framework for the Atacama Desert. TephAta provides easy access to complex information that normally is distributed in several media (publications, private or public databases), hence fostering FAIR

(findable, accessible, interoperable and reusable) data principals (Wilkinson et al., 2016). The design of TephAta is also meant to provide user friendly integration of suitable parts of the dataset (e.g. geochemical data) in global databases such as GEOROC (https://georoc.eu/) or the tephra-data-optimized EARTHCHEM (https://www.earthchem.org/) repository. This paper introduces the TephAta database and illustrates its usefulness by the presentation and discussion of an exemplary dataset obtained on widely distributed tephra deposits of Mid Pleistocene age from the central Atacama Desert.

## 2 The Central Volcanic Zone of the Andes

The subduction of the Farallon plate underneath the west coast of South America has caused continental arc volcanism within the Central Andes since Triassic times (Oliveros et al., 2018). The volcanic arc is characterized by low-flux, steady state andesitic magmatism and the construction of composite volcanoes (Burns et al., 2015). Starting about 25 Ma ago, an increase of the subduction rate and a shift in the subduction geometry of the Nazca Plate caused a eastward migration of the arc and associated crustal shortening and thickening (Allmendinger et al., 1997). This change is correlated with a shift to a high-flux, flare-up arc magmatism, which resulted in the production and eruption of large volumes of silicic "crustal" magmas (Wörner et al., 2018; Burns and De Silva, 2023). Ignimbrite volcanism is supposed to be not contemporaneous within the Central Volcanic Zone, indicating a north-south gradient towards younger major eruptive events (De Silva, 1989b; De Silva and Kay, 2018). The oldest ignimbrite deposits are incorporated within the Altos de Pica Formation (22-25 Ma; Jordan et al., 2014), the Oxaya Formation (19-23 Ma; Wörner et al., 2002; Van Zalinge et al., 2016) and the El Diablo Formations (11-16 Ma; Jordan et al., 2014). Between 10 and 1 Ma, magmatic fluxes strongly increased and numerous voluminous ignimbrites were erupted in pulses peaking at ca. 8.4 Ma, 5.5 Ma and 4.0 Ma (De Silva, 1989b; Sáez et al., 1999; Kay et al., 2010; Salisbury et al., 2011; Burns and De Silva, 2023). One of the most prominent surface morphologies of these ignimbrite deposits is the Altiplano-Puna Volcanic Complex (APVC) between 21 and 24°S (De Silva, 1989a). Subsequently, the high-flux volcanic activity waned and the construction of composite volcanoes and small-volume lava domes indicate the return to steady-state conditions (Burns and De Silva, 2023).

## 3 Database structure, samples, and analytical data

### 3.1 Database Structure

TephAta is hosted within the CRC1211 database (https://www.crc1211db.uni-koeln.de) and is focused entirely on storing and displaying information without data analysis by respective algorithms. In the backend, TephAta uses a relational database using MariaDB as data management system, communicating with a basic PHP instance hosted by the ITCC on virtual machines, while the frontend is handled by standard web technologies, with some display libraries (such as Tabulator or HighCharts) providing JavaScript interactivity. TephAta is designed to store site and sample metadata of tephra samples in combination with respective analytical datasets. The requested data structure is adapted to the global tephra community guidelines for data acquisition and reporting outlined in Wallace et al. (2022). Data can origin from new investigations, but also from published work.

TephAta is organized into seven data categories: site, sample, geochemistry, morphology, chronology, stratigraphy, and equivalent (Fig. 1a). A general introduction of the data stored in the different categories is given below and a full list and description of all input fields and upload functions is given in Supplementary Table 1.

The category "site" allows the documentation of sample location details including information about the site name (title), a general (log-) description, geographic position, physical habitus, and spatial context of an investigated location. The site category lists and links all samples taken from the same archive.

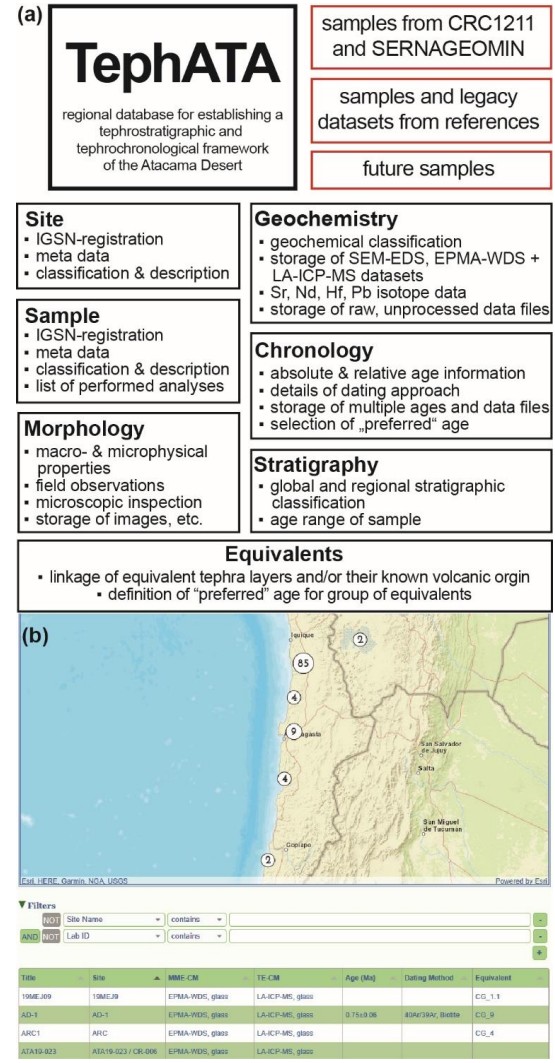

**Figure 1: S (a)** Structure of TephATA with its seven categories and different sources for data. **(b)** An example employing the sample filter function of TephATA showing the filtered samples on a map and listing them below (CRC1211-TephATA, 2025).

The section "sample" provides information when, how and by whom a sample from a specific site was taken. Further details about the sample type and material, its position within a site and the observed depositional processes can be defined. Information about sub- or split-samples and related (laboratory) labels can be listed together with the availability and accessibility of material and data. Specific analyses that were carried out can be indicated in an overview list.

The "geochemistry" data input allows definition of the volcanic rock type of a specific sample based on the total alkali vs silica (TAS) classification (Le Bas et al., 1986). Geochemical datasets of major (>1 wt.%), minor (1-0.1 wt.%) and trace (<0.1 wt.%) element data, but also Sr, Nd, Pb isotope data can be archived. Unprocessed raw data, instrument log files and laboratory protocol data of geochemical analyses can be stored as supplementary files.

The category "morphology" collects morphological and sample component observations from the field and from microscopic inspection. These include descriptions of the macrophysical properties of the stratigraphic layer where a specific sample was taken from (thickness, color, sedimentological structures, components), but also its microphysical properties (glass fragment morphology, mineral assemblages, degree of alteration). Field sketches, images or notes and microscopic images can also be integrated.



The "chronology" category collects age information related to a specific sample. Details on the applied dating technique(s) can be entered along with information about the dated material and specific age information (uncertainties, type of age calculation, xenocryst presence). Full analytical datasets including laboratory details as well as unprocessed and processed data-files (e.g., single crystal dating results) can be stored as supplementary files. Multiple ages (e.g., by different dating

techniques) can be entered for a single sample including the definition of a "best age", which will be assigned as the most reliable age in the sample overview.

Within the data category "stratigraphy" a specific sample can be categorized by its general (e.g. Erathem, Stage, etc.) and regional stratigraphic classifications. If (chrono-)stratigraphic information allows, a minimum and a maximum age can be defined.

For samples for which an equivalent tephra sample or volcanic eruption is identified and confirmed by tephrostratigraphic means, a link between samples can be documented within the "equivalent" category. Equivalent groups can be further defined by a common name and their most reliable age. In case a tephra layer can be traced back to its specific volcanic origin, the source volcano and eruption can be linked to the equivalent group.

All the aforementioned data is entered via the TephAta web interface (https://www.crc1211db.uni-koeln.de/tephata) using

guided input forms, including unified predefined lists, as well as free text fields for the input of individual data items. Geochemical data can be uploaded (and downloaded) by template data sheets provided on the website. The template files combine sample and instrument metadata (e.g., sample IDs and types, related references, applied analytical protocols and instrument) with respective results of standards and samples to ensure transparent documentation for interoperability and data reuse. For all categories additional data files (e.g., raw data and detailed instrument settings, field, and sample photo

documentation) can be uploaded as zip-files. Datasets stored within TephAta can be continuously extended, as additional analyses and methods become available. Based on the site and sample metadata, TephAta requests IGSNs (International Generic Sample Number), to enable a unique identification of a site and a sample. The IGSN also allows documentation outside TephAta and provides a connection to other databases. Data exchange to other global databases is also fostered by the structures of file templates, which are organized to match the needs of the global tephra-data repository EarthChem

(https://www.earthchem.org/; Kuehn et al., 2023).

Datasets stored within TephAta can be accessed through the categories: "sites" and "samples". The sample list is linked to a map tool, which allows exploring entries within a specific area (Fig. 1b). The sample list can be adapted by combinable filters, for specific site, sample and equivalent names (site name, sample name, lab ID, equivalent title), the type of site (e.g., only alluvial fans) and geochemical properties (TAS-classification, range of specific major, minor and/or trace element

concentrations). Samples can be also filtered for their stratigraphic (series) and chronological (age range) values. Furthermore, users can select up to 15 samples to create compositional x-y plots. An overview of established tephra correlations is given by the "equivalent groups" list. These functions shall foster the use of the database to identify potential equivalent tephra layers and thus provide the basis for detailed testing and subsequent establishment of tephrostratigraphic and -chronological correlations.



**Table 1**: Overview table listing all sites and samples included within TephATA along with information about their type of site, proposed age and dating details (if available, age and age uncertainty, dating method, dated mineral, dating laboratory, mineral standard and decay constant applied, type of age and reference of age). Further it is listed if the tephra was previously known and the sample was provided or (re-)sampled within the activities of the CRC1211, more details about the origin of samples are given within TephAta database. Sample AD-85 was provided by J. Quade (University of Arizona) and sample

TJ09-PdT-1 was provided by T. Jordan (Cornell University).

References are:

1=Breitkreuz et al. (2014)
3 = Sáez et al. (2012)
4= Ritter et al. (2018)
5= this study
6= Gardeweg and Sellés (2013)
7= Astudillo et al. (2017)

8= Blanco and Tomlinson (2013)
9= Medina et al. (2012)
10=Escribano et al. (2013)
11= Marinovic et al. (1995)
12= Carrizo et al. (2008)
13=Vásquez et al. (2018)
14=Vásquez and Sepúlveda (2013)
15= Sepúlveda et al. (2014)

16= Rodríguez et al. (2015)
17= Horn (1991)
18=Wörner et al. (2000)
19= Medialdea et al. (2020)
20= Fornari et al. (2001)
21= Fritz et al. (2004)
22= Sepúlveda et al. (2023)
23= May et al. (2020)

2= Placzek et al. (2009)

| site | sample | 2nd sample name / resampled equivalent | type of site | age (Ma) | age uncertainty (Ma) | dating method | dated mineral | dating laboratory | Ar-Ar mineral standard, decay constant | type of age | reference | sample source | known or new sample |
|---|---|---|---|---|---|---|---|---|---|---|---|---|---|
| 10-3-5-1 | 10-3-5-1 | L-10 | alluvial fan | not dated | not dated | N/A | N/A | N/A | N/A | N/A | 1 | reference | known |
| 19MEJ9 | 19MEJ09 | - | alluvial fan | not dated | not dated | N/A | N/A | N/A | N/A | N/A | - | CRC | new |
| AD-1 | AD-1 | - | alluvial fan | 0.750 | 0.060 | $^{40}Ar/^{39}Ar$ | biotite | Sernageomin | FC; Steiger & Jäger (1977) | plateau | 2 | reference | known |
| ARC | ARC1 | - | alluvial fan | not dated | not dated | N/A | N/A | N/A | N/A | N/A | 3 | reference | known |
| Asche 4 / PAG-T4 | PAG-T4 | PAG TEPHRA 4 | channel | 0.980 | 0.040 | U/Pb | zircon | University of Frankfurt | N/A | TuffZirc | 4 | CRC | new |
| Asche 4 / PAG-T4 | PAG-T4 | PAG TEPHRA 4 | channel | 0.301 | 0.007 | $^{40}Ar/^{39}Ar$ | biotite | University of Amsterdam | FC; Min et al. (2000) | weighted mean total fusion | 5 | CRC | new |
| ATA19-023 / CR-006 | ATA19-023 | CR-006A | ignimbrite deposits | equivalent dated | equivalent dated | N/A | N/A | N/A | N/A | N/A | - | CRC | known |
| ATA19-023 / CR-006 | CR-006A | ATA19-023 | ignimbrite deposits | 0.280 | 0.014 | $^{40}Ar/^{39}Ar$ | biotite | Sernageomin | FC; Steiger & Jäger (1977) | plateau | 6 | reference | known |
| AVN-002 | AVN-002d | - | alluvial fan | 0.740 | 0.080 | $^{40}Ar/^{39}Ar$ | biotite | Sernageomin | FC; Steiger & Jäger (1977) | plateau | 7 | SERNAGEOMIN | known |
| C2-TEP-04/IT93 | C2-TEP-04 | IT-93 | channel | equivalent dated | equivalent dated | N/A | N/A | N/A | N/A | N/A | - | CRC | known |
| C2-TEP-04/IT93 | IT-93 | C2-TEP-04 | channel | 0.170 | 0.040 | $^{40}Ar/^{39}Ar$ | biotite | Sernageomin | FC; Steiger & Jäger (1977) | plateau | 8 | reference | known |
| C2-TEP-05/IT307 | C2-TEP-05 | IT-307 | channel | equivalent dated | equivalent dated | N/A | N/A | N/A | N/A | N/A | - | CRC | known |
| C2-TEP-05/IT307 | IT-307 | C2-TEP-05 | channel | 0.400 | 0.100 | K-Ar | biotite | Sernageomin | Steiger & Jäger (1977) | N/A | 8 | reference | known |

| site | sample | 2nd sample name / resampled equivalent | type of site | age (Ma) | age uncertainty (Ma) | dating method | dated mineral | dating laboratory | Ar-Ar mineral standard, decay constant | type of age | ref-erence | sample source | known or new sample |
|---|---|---|---|---|---|---|---|---|---|---|---|---|---|
| CC157 / L8 / 2/3/5/2 | 2/3/5/2 | CC-157 // L8 | alluvial fan | equivalent dated | equivalent dated | N/A | N/A | N/A | N/A | N/A | 1 | reference | known |
| CC157 / L8 / 2/3/5/2 | CC-157 | 2/3/5/2 // L8 | alluvial fan | 0.230 | 0.130 | $^{40}Ar/^{39}Ar$ | biotite | Sernageomin | FC; Steiger & Jäger (1977) | inverse isochron | 9 | reference | known |
| CH17-001 (AD-85) | AD-85 | CH17-001 | channel | not dated | not dated | N/A | N/A | N/A | N/A | N/A | - | J. Quade | new |
| CH17-001 (AD-85) | CH17-001 | AD-85 | channel | not dated | not dated | N/A | N/A | N/A | N/A | N/A | - | CRC | new |
| CH18-T1-4 | CH18-T1 | - | alluvial fan | not dated | not dated | N/A | N/A | N/A | N/A | N/A | - | CRC | new |
| CH18-T1-4 | CH18-T2 | - | alluvial fan | not dated | not dated | N/A | N/A | N/A | N/A | N/A | - | CRC | new |
| CH18-T1-4 | CH18-T3 | - | alluvial fan | not dated | not dated | N/A | N/A | N/A | N/A | N/A | - | CRC | new |
| CH18-T1-4 | CH18-T4 | - | alluvial fan | not dated | not dated | N/A | N/A | N/A | N/A | N/A | - | CRC | new |
| CH18-T5 | CH18-T5 | - | alluvial fan | not dated | not dated | N/A | N/A | N/A | N/A | N/A | - | CRC | new |
| CH22-NL-T1 | CH22-NL-T1 | TPM-094 | alluvial fan | 3.900 | 0.800 | $^{40}Ar/^{39}Ar$ | biotite | Sernageomin | FC; Steiger & Jäger (1977) | combined isochron n=2 | 10 | CRC | known |
| CH22-NL-T2 | CH22-NL-T2 | MAB-663 | alluvial fan | 5.500 | 1.000 | K-Ar | biotite | Sernageomin | N/A | N/A | 11 | CRC | known |
| CH22-NL-T3 | CH22-NL-T3-1 | - | alluvial fan | not dated | not dated | N/A | N/A | N/A | N/A | N/A | - | CRC | known |
| CH22-NL-T3 | CH22-NL-T3-2 | - | alluvial fan | not dated | not dated | N/A | N/A | N/A | N/A | N/A | - | CRC | known |
| CHU / AN1 | CHU Tephra | AN1 | fault scarp | equivalent dated | equivalent dated | N/A | N/A | N/A | N/A | N/A | - | SERNAGEOMIN | known |
| CHU / AN1 | AN-1 | CHU-Tephra | fault scarp | 0.310 | 0.190 | $^{40}Ar/^{39}Ar$ | biotite | Sernageomin | N/A | plateau | 12 | reference | known |
| CHU-1 | CHU-1 | - | channel | not dated | not dated | not dated | N/A | N/A | N/A | N/A | - | SERNAGEOMIN | known |
| CHU-2 | CHU-2 | - | badlands | not dated | not dated | not dated | N/A | N/A | N/A | N/A | - | SERNAGEOMIN | known |
| CON-1 | CON-1 | - | channel | 0.181 | 0.056 | $^{40}Ar/^{39}Ar$ | biotite | GEOMAR Kiel | TCR-2, unknown | plateau | 3 | reference | known |
| CTFON-4 | CTFON-4 | - | marine deposits | not dated | not dated | not dated | N/A | N/A | N/A | N/A | - | CRC | known |
| EL Rincon | 19MEJ10 | - | marine deposits | not dated | not dated | N/A | N/A | N/A | N/A | N/A | - | CRC | new |
| EL Rincon | 23MEJ1 | - | marine deposits | not dated | not dated | N/A | N/A | N/A | N/A | N/A | - | CRC | new |
| ESQ-1 | ESQ-1 | - | channel | 0.151 | 0.033 | $^{40}Ar/^{39}Ar$ | biotite | GEOMAR Kiel | TCR-2 | plateau | 3 | reference | known |
| GSQ-027 | GSQ-027d | - | terrace | 0.255 | 0.017 | $^{40}Ar/^{39}Ar$ | biotite | Sernageomin | FC; Steiger & Jäger (1977) | plateau | 13 | SERNAGEOMIN | known |
| GSQ-08 | GSQ-08d | - | terrace | 0.300 | 0.020 | $^{40}Ar/^{39}Ar$ | biotite | Sernageomin | FC; Steiger & Jäger (1977) | plateau | 13 | SERNAGEOMIN | known |
| GSQ-106d | GSQ-106d | - | alluvial fan | 0.329 | 0.029 | $^{40}Ar/^{39}Ar$ | biotite | Sernageomin | FC; Steiger & Jäger (1977) | plateau | 13 | SERNAGEOMIN | known |
| GSQ-158 | GSQ-158d | - | alluvial fan | 0.340 | 0.021 | $^{40}Ar/^{39}Ar$ | biotite | Sernageomin | FC; Steiger & Jäger (1977) | plateau | 13 | SERNAGEOMIN | known |
| GSQ-53 | GSQ-53d | - | alluvial fan | 0.325 | 0.048 | $^{40}Ar/^{39}Ar$ | biotite | Sernageomin | FC; Steiger & Jäger (1977) | plateau | 13 | SERNAGEOMIN | known |





| site | sample | 2nd sample name / resampled equivalent | type of site | age (Ma) | age uncertainty (Ma) | dating method | dated mineral | dating laboratory | Ar-Ar mineral standard, decay constant | type of age | ref-erence | sample source | known or new sample |
|---|---|---|---|---|---|---|---|---|---|---|---|---|---|
| GSV-132 | GSV-132d | - | fault scarp | 0.140 | 0.057 | ⁴⁰Ar/³⁹Ar | biotite | Sernageomin | FC; Steiger & Jäger (1977) | plateau | 13 | SERNAGEOMIN | known |
| HU14/008 (IS-155) | HU14/008 | IS-155 | claypan | equivalent dated | equivalent dated | N/A | N/A | N/A | N/A | N/A | - | CRC | known |
| HU14/008 (IS-155) | IS-155 | HU14/008 | claypan | 22.9 | 0.3 | ⁴⁰Ar/³⁹Ar | biotite | Sernageomin | FC; Steiger & Jäger (1977) | plateau | 14 | SERNAGEOMIN | known |
| HU17-014 | HU17-014 | - | channel | not dated | not dated | N/A | N/A | N/A | N/A | N/A | - | CRC | new |
| HU18/001/002 | HU18/001 | - | channel | not dated | not dated | N/A | N/A | N/A | N/A | N/A | - | CRC | new |
| HU18/001/002 | HU18/002 | - | channel | not dated | not dated | N/A | N/A | N/A | N/A | N/A | - | CRC | new |
| HU18-003 (IV-190) | HU18/003 | IV-190 | channel | equivalent dated | equivalent dated | N/A | N/A | N/A | N/A | N/A | - | CRC | known |
| HU18-003 (IV-190) | IV-190 | HU18/003 | channel | 0.620 | 0.040 | ⁴⁰Ar/³⁹Ar | biotite | Sernageomin | FC; Steiger & Jäger (1977) | plateau | 14 | SERNAGEOMIN | known |
| HU18-008 (IV-189) | HU18/008 | IV-189 | channel | equivalent dated | equivalent dated | N/A | N/A | N/A | N/A | N/A | - | CRC | known |
| HU18-008 (IV-189) | IV-189 | HU18-008 | channel | 2.06 | 0.12 | ⁴⁰Ar/³⁹Ar | biotite | Sernageomin | FC; Steiger & Jäger (1977) | inverse isochron | 14 | reference | known |
| IA-05 | IA-05 | - | alluvial fan | 0.40 | 0.20 | K-Ar | biotite | Sernageomin | FC; Steiger & Jäger (1977) | combined K-Ar n=2 | 15 | SERNAGEOMIN | known |
| IA-105 | IA-105 | - | alluvial fan | 0.314 | 0.012 | ⁴⁰Ar/³⁹Ar | biotite | Sernageomin | FC; Steiger & Jäger (1977) | plateau | 15 | SERNAGEOMIN | known |
| IA-11 | IA-11 | - | channel | 0.430 | 0.040 | ⁴⁰Ar/³⁹Ar | biotite | Sernageomin | FC; Steiger & Jäger (1977) | inverse isochron | 15 | SERNAGEOMIN | known |
| IA-120 | IA-120 | - | alluvial fan | 0.300 | 0.200 | K-Ar | biotite | Sernageomin | FC; Steiger & Jäger (1977) | combined K-Ar n=2 | 15 | SERNAGEOMIN | known |
| IA-137 | IA-137 | - | channel | 0.700 | 0.400 | K-Ar | biotite | Sernageomin | FC; Steiger & Jäger (1977) | N/A | 15 | SERNAGEOMIN | known |
| IA-162 | IA-162 | - | alluvial fan | 0.900 | 0.200 | K-Ar | biotite | Sernageomin | FC; Steiger & Jäger (1977) | combined K-Ar n=2 | 15 | SERNAGEOMIN | known |
| CH22-NL-T46 | CH22-NL-T46 | IA-270? | hillslope | equivalent dated | equivalent dated | N/A | N/A | N/A | N/A | N/A | - | CRC | known |
| IA-270 | IA-270 | CH22-NL-T46? | alluvial fan | 0.410 | 0.030 | ⁴⁰Ar/³⁹Ar | biotite | Sernageomin | FC; Steiger & Jäger (1977) | plateau | 15 | SERNAGEOMIN | known |
| IA-95 | IA-95 | - | alluvial fan | 0.500 | 0.300 | K-Ar | biotite | Sernageomin | FC; Steiger & Jäger (1977) | N/A | 15 | SERNAGEOMIN | known |
| IA-97 | IA-97 | - | alluvial fan | 0.600 | 0.300 | K-Ar | biotite | Sernageomin | FC; Steiger & Jäger (1977) | combined K-Ar n=2 | 15 | SERNAGEOMIN | known |
| IQ18-001 | IQ18-001 | - | alluvial fan | not dated | not dated | N/A | N/A | N/A | N/A | N/A | - | CRC | known |



| site | sample | 2nd sample name / resampled equivalent | type of site | age (Ma) | age uncertainty (Ma) | dating method | dated mineral | dating laboratory | Ar-Ar mineral standard, decay constant | type of age | ref-erence | sample source | known or new sample |
|---|---|---|---|---|---|---|---|---|---|---|---|---|---|
| IQ18-002 (CMI13.3) | CMI13.3 | IQ18-002 | alluvial fan | 0.700 | 0.110 | $^{40}$Ar/$^{39}$Ar | biotite | Sernageomin | FC; Steiger & Jäger (1977) | plateau | 14 | SERNAGEOMIN | known |
| IQ18-002 (CMI13.3) | IQ18-002 | CMI13.3 | alluvial fan | equivalent dated | equivalent dated | N/A | N/A | N/A | N/A | N/A | - | CRC | known |
| IR22-01 | IR22-001 | - | ignimbrite deposits | not dated | not dated | N/A | N/A | N/A | N/A | N/A | - | CRC | known |
| IR22-02 | IR22-002 | - | ignimbrite deposits | not dated | not dated | N/A | N/A | N/A | N/A | N/A | - | CRC | known |
| IRRU-13 | IRRU-13 | - | flank of volcano | not dated | not dated | N/A | N/A | N/A | N/A | N/A | 16 | reference | known |
| IRRU-46a | IRRU-46a | - | flank of volcano | 0.258 | 0.049 | $^{40}$Ar/$^{39}$Ar | biotite | Oregon State University | FC | inverse isochron | 16 | reference | known |
| IRU-15 | IRU-15 | - | ignimbrite deposits | 0.320 | 0.250 | K-Ar | biotite | NERC | N/A | plateau | 17, 8 | reference | known |
| IRU-1a | IRU-1a | - | flank of volcano | 0.181 | 0.009 | $^{40}$Ar/$^{39}$Ar | biotite | University of Amsterdam | FC; Min et al. (2000) | weighted mean total fusion | 5, 17, 18 | reference | known |
| PAG | PAG17 ID14 | - | claypan | not dated | not dated | N/A | N/A | N/A | N/A | N/A | - | CRC | new |
| PAG | PAG17 ID24 | - | claypan | not dated | not dated | N/A | N/A | N/A | N/A | N/A | - | CRC | new |
| PAG | PAG17 ID4 | - | claypan | not dated | not dated | N/A | N/A | N/A | N/A | N/A | - | CRC | new |
| PAG | PAG17 ID54 | - | claypan | not dated | not dated | N/A | N/A | N/A | N/A | N/A | - | CRC | new |
| PAG T-1-2 | PAG-T1 | PAG TEPHRA 1 | claypan | not dated | not dated | N/A | N/A | N/A | N/A | N/A | - | CRC | new |
| PAG T-1-2 | PAG-T2 | PAG TEPHRA 2 | channel | not dated | not dated | N/A | N/A | N/A | N/A | N/A | - | CRC | new |
| PAG17.2 008 | PAG17.2-008 | - | channel | not dated | not dated | N/A | N/A | N/A | N/A | N/A | - | CRC | new |
| PAG17/001 | PAG17/001 | - | alluvial fan | not dated | not dated | N/A | N/A | N/A | N/A | N/A | - | CRC | new |
| PAG17/006 | PAG17/006 | - | channel | not dated | not dated | N/A | N/A | N/A | N/A | N/A | - | CRC | new |
| PAG17/027 | PAG17/027 | - | channel | not dated | not dated | N/A | N/A | N/A | N/A | N/A | - | CRC | new |
| PAG18/001 | PAG18/001 | - | channel | not dated | not dated | N/A | N/A | N/A | N/A | N/A | - | CRC | new |
| PAG18/002 | PAG18/002 | - | alluvial fan | not dated | not dated | N/A | N/A | N/A | N/A | N/A | - | CRC | new |
| PAG18/003 | PAG18/003 | - | alluvial fan | not dated | not dated | N/A | N/A | N/A | N/A | N/A | - | CRC | new |
| PAG18/008 | PAG18/008 | - | channel | not dated | not dated | N/A | N/A | N/A | N/A | N/A | - | CRC | new |
| POR 3 | POR 3 | - | flank of volcano | 0.290 | 0.450/0.290 | K-Ar | feldspar | NERC | Steiger and Jager, 1977 | No reliable ages | 17, 18 | reference | known |
| POR 3 | POR 3 | - | flank of volcano | 0.150 | 0.200/0.150 | K-Ar | biotite | NERC | Steiger and Jager, 1977 | No reliable ages | 17, 18 | reference | known |
| QP | QP1 | - | channel | not dated | not dated | N/A | N/A | N/A | N/A | N/A | 3 | reference | known |
| QP | QP4 | - | channel | not dated | not dated | N/A | N/A | N/A | N/A | N/A | 3 | reference | known |
| Quebrada Ancha | Tephra RB | - | channel | not dated | not dated | N/A | N/A | N/A | N/A | N/A | - | CRC | new |
| Quebrada Tiburón - Section A | TIB4-TEPH1 | - | marine deposits | not dated | not dated | N/A | N/A | N/A | N/A | N/A | - | CRC | new |





| site | sample | 2nd sample name / resampled equivalent | type of site | age (Ma) | age uncertainty (Ma) | dating method | dated mineral | dating laboratory | Ar-Ar mineral standard, decay constant | type of age | ref-erence | sample source | known or new sample |
|---|---|---|---|---|---|---|---|---|---|---|---|---|---|
| Quebrada Tiburón - Section A | TIB4-F2 | - | marine deposits | not dated | not dated | N/A | N/A | N/A | N/A | N/A | | CRC | new |
| RL North | Tephra RL N | - | alluvial fan | not dated | not dated | N/A | N/A | | N/A | N/A | - | CRC | new |
| SALAR | SG17/001 | SALAR T6 | hillslope | equivalent dated | equivalent dated | N/A | N/A | N/A | N/A | N/A | - | CRC | new |
| SALAR | SG17/002 | SALAR T4 | hillslope | equivalent dated | equivalent dated | N/A | N/A | N/A | N/A | N/A | - | CRC | new |
| SALAR | SG17/003 | SALAR T2 | hillslope | equivalent dated | equivalent dated | N/A | N/A | N/A | N/A | N/A | - | CRC | new |
| SALAR | SALAR T6 | SG17/001 | hillslope | 1.297 | 0.018 | $^{40}Ar/^{39}Ar$ | sanidine | University of Amsterdam | FC; Min et al., 2000 | weighted mean total fusion | 19 | CRC | new |
| SALAR | SALAR T6 | - | hillslope | 0.098 | 0.015 | indirect - sediment OSL dating + absolute | N/A | University of Cologne | N/A | AGE-MODEL | 19 | CRC | new |
| SALAR | SALAR T5 | - | hillslope | 0.084 | 0.008 | indirect - sediment OSL dating | N/A | University of Cologne | N/A | AGE-MODEL | 19 | CRC | new |
| SALAR | SALAR T4 | SG17/002 | hillslope | 0.083 | 0.009 | indirect - sediment OSL dating | N/A | University of Cologne | N/A | AGE-MODEL | 19 | CRC | new |
| SALAR | SALAR T3 | - | hillslope | 0.072 | 0.008 | indirect - sediment OSL dating | N/A | University of Cologne | N/A | AGE-MODEL | 19 | CRC | new |
| SALAR | SALAR T2 | SG17/003 | hillslope | 0.071 | 0.007 | indirect - sediment OSL dating | N/A | University of Cologne | N/A | AGE-MODEL | 19 | CRC | new |
| SALAR | SALAR T1 | - | hillslope | 0.068 | 0.008 | indirect - sediment OSL dating | N/A | University of Cologne | N/A | AGE-MODEL | 19 | CRC | new |
| Salar de Uyuni - C1986 | L5 | TSdU | salar | 0.191 | 0.005 | $^{40}Ar/^{39}Ar$ | biotite | Université de Nice — Sophia-Antipolis | Bern 4B biotite, N/A | fusion + isochron | 20 | reference | known |
| Salar de Uyuni - C1999 | TSdU | L5 | salar | 0.059 | n/a | indirect U/Th of host sediment | halite | University of California? | N/A | AGE-MODEL | 21 | reference | known |
| SMS-15d | SMS-15d | - | channel | 0.375 | 0.017 | $^{40}Ar/^{39}Ar$ | biotite | Sernageomin | FC; Steiger & Jäger (1977) | plateau | 22 | SERNAGEOMIN | known |
| T3 | SGL3 | - | alluvial fan | 0.080 | n/a | indirect - sediment OSL dating | N/A | University of Cologne | N/A | AGE-MODEL | 23 | CRC | new |
| TA-7-8 | TA-8 | - | channel | 0.210 | 0.066 | $^{40}Ar/^{39}Ar$ | biotite | GEOMAR Kiel | TCR-2 | plateau | 3 | reference | known |
| TA-7-8 | TA-7 | - | channel | 0.098 | 0.042 | $^{40}Ar/^{39}Ar$ | biotite | GEOMAR Kiel | TCR-2 | plateau | 3 | reference | known |
| TJ09-PdT1 | TJ09-PdT1 | - | playa | not dated | not dated | N/A | N/A | N/A | N/A | N/A | - | T. Jordan | new |
| TOC14/001 | TOC14/001 | - | alluvial fan | not dated | not dated | N/A | N/A | N/A | N/A | N/A | - | CRC | new |




3.2 Samples and chronological and geochemical data

TephAta sets a focus on tephra samples from the Atacama Desert and adjacent regions, namely the Coastal Cordillera, the
Central Depression, Precordillera and parts of the Western Cordillera independent of their age. Until now, mainly tephra
samples were included, which are assigned to the late Middle Pleistocene to Late Pleistocene and were collected mainly during
field campaigns within the CRC1211 (2013-2023). Samples are derived from sites, which were within the scientific focus
areas of CRC1211 subprojects and from adjacent sites to create a broad data foundation for developing a tephrostratigraphic
framework and to characterize major eruptive events. This set of samples was augmented by identifying tephra sites on
geological maps and/or described in literature. In case that tephra layers were identified in the vicinity of an investigated
CRC1211 archive or of similar age, sample splits were requested from the Chilean National Geology and Mining Survey
(SERNAGEOMIN) or the respective researchers of literature sources. If material could not be provided, respective sample
sites were tried to be revisited and resampled during field campaigns of the CRC1211. TephAta also allows the integration of
legacy data reported in the literature for samples which are physically not available anymore. The samples, which are currently
investigated, have a heterogeneous origin from different projects and decades and related metadata and field observations were
originally not collected based on a standardized protocol. TephAta has been designed to accommodate the diverse range of
data types within a single, unified framework. This is achieved by offering a comprehensive set of optional data fields, which
can be utilized if needed and data is available. Metadata and morphological field descriptions of sites and samples were
extracted from references and field notes or provided by personal communication.


Chronological data

Available ages for the samples currently included in TephATA have been extracted from published data and were
complemented by new chronological results. Available ages from literature were obtained by $^{40}Ar/^{39}Ar$, K-Ar and U-Pb dating,
but also inferring indirect tephra ages by dating the host sediment. Available details of the respective dating techniques, age
calculations and references are stored sample-specific within TephAta and are summarized in Table 1.

For two samples (PAG-T4 and IRU-1a) new ages were obtained by $^{40}Ar/^{39}Ar$ multi-grain fusion dating of biotite at the Vrije
Universiteit Amsterdam and are presented here for the first time. Mineral separates were achieved by sieving, magnetic
separation, and hand picking before the selected mineral separate was packed in a 6 mm ID Al packages and loaded in a 25
mm ID Al cup together with Fish Canyon Tuff sanidine (FCs) standard. Sample and standard were irradiated at the Oregon
State University TRIGA reactor in the cadmium shielded CLICIT facility for 7 hours (irradiation code VU114 and 15-OSU-
06). At Vrije Universiteit Amsterdam after irradiation, samples and standards were unpacked and loaded in a 185 hole Cu tray
and baked overnight at 250 °C under vacuum. This tray was then placed in a doubly pumped vacuum chamber with Zn-Se
window and baked overnight at 120 °C under high vacuum. This chamber is connected to a ThermoFisher NGPrep gas
purification line equipped with a hot GP50, a cold finger (Lauda at -70 °C) and hot St707 getter. Samples (1-3 grains/fusion)
and standards (1 grain/fusion) are fused using a 25 W Synrad $CO_2$ laser. Released gas is analyzed on an ARGUS VI+ noble
gas mass spectrometer, equipped with four Faraday cups at the H2, H1, AX and L1 positions and two compact discrete dynodes
(CDDs) at positions L2 and L3. The system is equipped with a 1012 Ohm amplifier on H2 and 1013 Ohm amplifiers on H1,
AX and L1 cups. Samples were run on H1-L3 collectors. Similar to Phillips and Matchan (2013), no bias corrections were
applied, but samples and standards were analyzed in the same tray (and thus at more or less the same time) alternating with air
pipettes with intensities in the same range as the samples and standards. Line blanks were measured every 2-3 unknowns and
were subtracted from succeeding sample data. Data reduction is done in ArArCalc (Koppers, 2002). Ages are calculated with
decay constants of Min et al. (2000) and 28.201 Ma for FCs (Kuiper et al., 2008). The atmospheric $^{40}Ar/^{36}Ar$ air value of
298.56 is used (Lee et al., 2006). The correction factors for neutron interference reactions are $(2.64 \pm 0.02) \times 10^{-4}$ for $(^{36}Ar/^{37}Ar)$
Ca, $(6.73 \pm 0.04) \times 10^{-4}$ for $(^{39}Ar/^{37}Ar)$ Ca, $(1.21 \pm 0.003) \times 10^{-2}$ for $(^{38}Ar/^{39}Ar)$ K and $(8.6 \pm 0.7) \times 10^{-4}$ for $(^{40}Ar/^{39}Ar)$ K. All
errors are quoted at the 2σ level.



Geochemical data

Depending on individual sample properties, different preparation methods were applied to enrich volcanic glass fragments for geochemical analyses. These methods include classical preparation techniques, like crushing of lithified/solidified samples, rinsing, (wet-) sieving, magnetic and density separation, and hand picking. Specific information for each sample is listed in

the sample description and is also given along with the analytical results within the data-files provided for download (e.g., results of geochemical EPMA-WDS analyses). Respective aliquot fractions enriched in glass fragments were mounted on epoxy pucks and polished to remove potential surficial alteration and to avoid topographic effects causing compositional variations during subsequent glass geochemical analyses.

For electron microprobe wavelength dispersive spectroscopy (EPMA-WDS) and scanning electron microscope energy

dispersive spectroscopy (SEM-EDS) the pucks were carbon coated and analyzed for their major and minor element glass composition. A JEOL JXA-8900RL electron microprobe equipped with five-wavelength dispersive spectrometers was used for these analyses at the University of Cologne (UoC). The operation conditions were set to 12 kV accelerating voltage, 6 nA beam current and 5 μm beam diameter. Full details of calibration and measuring conditions are given in Leicher (2021). Samples indicating the presence of multiple geochemical populations or trends of magmatic evolution within the EPMA-WDS

glass-composition data were further analyzed by SEM-EDS, if subject to LA-ICP-MS trace element analyses. For these samples, the same glass fragments analyzed by LA-ICP-MS were investigated for their major and minor element composition. The focus was to obtain grain-specific $SiO_2$ concentrations, which were then used as internal standard during LA-ICP-MS trace element data reduction. At the UoC, a Zeiss Sigma 300-VP equipped with an OXFORD Instruments EDX detector controlled by the software Aztec 4.1 was used for SEM-EDS analyses. The instrument was set to 20 keV and an image

resolution of 512*512 pixel was chosen. The number of channels was set to 2048, processing time to 5, with a dwell time of 10000 μs. The size of the scanning frame or line was adjusted to the respective grain size and typically exceeded 10 μm.

Trace-element analyses of glass fragments were performed at the University of Bonn (UoB) and UoC on the same sample pucks as the EPMA-WDS/SEM-EDS analyses, after removing the carbon coating. At the UoB trace element analyses were performed using a Resonetics Resolution M50E 193 nm excimer laser ablation system coupled to a Thermo Scientific Element

XR (Leicher and Lagos, 2021). A spot size of 26 μm was chosen and analyses were performed at a repetition rate of 5 Hz and a count time of 35 sec on the sample after 30 sec on the gas blank (background). A He gas flow (0.75 l min$^{-1}$), mixed together with Ar sample gas (~1.1 l min$^{-1}$) transported ablated material via an in-house signal-smoothing device to the ICP-MS. Maximum intensity as well as stability of the signal were obtained by tuning while taking account of concurrently low oxide ratios (ThO/Th of ~0.0012) to minimize potentially interfering oxide species prior to analyses in low-resolution mode. At UoC

an NWRimageGEO 193 nm ArF excimer laser ablation system coupled to a Thermo Scientific iCAP quadrupole or iCAP triple quadrupole ICP-MS was used for trace-element analyses. Individual glass fragments were analyzed using 15 or 20 μm spot size, a repetition rate of 6 Hz and a count time of 40 sec on the sample after 30 sec of the gas blank (background) measurements. The sample aerosol was taken up by a He gas flow (0.9 l min$^{-1}$) and mixed with Ar (0.8 l min$^{-1}$), within a glass smoothing device before introduction into the plasma.

All data reduction was performed using the software Iolite 4.3 (Paton et al., 2011). The standard glass NIST SRM 612 was used for calibration and calculation of trace element concentrations base on $^{43}Ca$, $^{85}Rb$, $^{88}Sr$, $^{89}Y$, $^{90}Zr$, $^{93}Nb$, $^{138}Ba$, $^{139}La$, $^{140}Ce$, $^{141}Pr$, $^{146}Nd$, $^{147}Sm$, $^{153}Eu$, $^{157}Gd$, $^{159}Tb$, $^{163}Dy$, $^{165}Ho$, $^{166}Er$, $^{169}Tm$, $^{172}Yb$, $^{175}Lu$, $^{178}Hf$, $^{181}Ta$, $^{208}Pb$, $^{232}Th$, $^{238}U$ and incorporating $^{29}Si$ as an internal standard. Si concentrations were obtained by EPMA-WDS or SEM-EDS and either considered as median values for samples with a narrow and homogenous silica composition or as grain specific values for tephra samples with

heterogeneous silica composition.

The here presented geochemical data can be either downloaded as sample-specific datasets from the TephATA database (https://www.crc1211db.uni-koeln.de/tephata) or can be found as a combined dataset of all samples (Leicher N., 2027) deposited at EarthChem.





Data quality of geochemical analyses

During EPMA-WDS, SEM-EDS and LA-ICP-MS analysis MPI-DING standard glasses (ATHO-G, StHs6/80-G; Jochum et al., 2006) and SRM NIST SRM 610 (Jochum et al., 2011) were used as secondary standards to evaluate the accuracy and precision of individual measurement sessions. Sample-specific results of secondary standard analysis are stored within TephAta along with the sample results and are provided for download within the sample-specific data-file.

The overall timeframe of EPMA-WDS analyses performed at UoC reveals mean values for intermediate precision (relative

standard deviation %) and accuracy (bias of mean to preferred reference value in %) of up to 1.9 and 0.9 %, respectively, for elemental concentrations >75 wt.%, up to 2.1 % and 1.1 % for 75-12 wt.%, up to 7.4 % and 1.2 % for 12-1 wt.% and up to 29.0 % and 5.9 % for <1 wt.%. Mean values for intermediate precision and accuracy of SEM-EDS analyses, respectively, are respectively both up to 0.3% for elemental concentrations > 75 wt.%, up to 1.9 % and 0.3 % for 75-12 wt.%, up to 9.0 % and 3.7 % for 12-1 wt.% and up to 30.0 % and 7. 9 % for <1 wt. %. For the currently implemented legacy glass EPMA-WDS data,

evaluation of data quality is hampered by the lack of reported results of secondary reference materials (Placzek et al., 2009; Breitkreuz et al., 2014). The respective legacy analyses stored in TephAta are flagged accordingly.

Secondary standard analyses of MPI-DING glasses ATHO-G and StHs6/80-G during LA-ICP-MS analyses at the UoB revealed median intermediate accuracies of <5 % for Sm, Ho, Rb, Ba, Pr, Ca, Sr, Ce, Yb, Lu, Hf, and Er, 5-12 % for La, Dy, Nd, Tm, U, Tb, Th, Eu, Y, Zr, Gd, Nb and Ta and 19 % for Pb. The median intermediate precision of the secondary standard

analyses at UoB was typically <5 % for Sr, Zr, Ce, Nb, Rb, Ba, La, Ca, Pr, Y, and Th and 5-14 % for Nd, U, Hf, Er, Ho, Pb, Dy, Eu, Gd, Ta, Sm, Tb, Lu, Yb, and Tm. At the UoC, secondary standard analyses of MPI-DING glasses (ATHO-G, StHs6/80-G) glasses during LA-ICP-MS analyses revealed median accuracies of 5-10 % for Rb, Sr, Ba, Ca, Ce, La, Sm, Ho, and U, 10-15 % for Zr, Nd, Th, Hf, Y, Pr, Eu, Tb, Er, Yb, Nb, Dy Pb and 15-17 % for Gd, Lu, Tm and Ta. The median intermediate precision of the secondary standard analyses at UoC was typically < 5 % for Sr, Ba, Zr, Ce, La, Rb, Nb, Y, Ca, Pr, Nd, Th, Hf,

Yb, U, Pb, Er, and Ta and of 5-8 % for Ho, Gd, Sm, Dy, Tb, Eu, Lu, and Tm.

## 4 Results and Discussion

### 4.1 Site and sample data

Currently, the TephATA dataset lists 106 tephra samples from 91 tephra deposits sampled at 74 sites between approximately 20 – 27° S to 70-67°W. The specific origin (locality, reference) of samples is listed in Table 1. The set of samples includes 65

samples taken previous to the CRC-research activities of which 23 samples originate from mapping campaigns of the SERNAGEOMIN and 25 samples from other regional geoscientific studies. In addition, 17 previously known sample sites were revisited and re-sampled as part of the CRC's field work during which also 41 newly identified tephra samples were gathered and included here.

Tephra samples are sourced from all type of sediment deposits of the Atacama Desert including alluvial fans, river terraces as

well as clay pan and halite deposits. In addition, a small set of samples originating from the proximal deposits associated with the Irruputuncu volcanic complex were included, because they represent a potential volcanic source. Sampling sites are mostly outcrops formed by the incision of channels, gravimetric movement, tectonic activity, and roadcuts, and also include pits and drill cores. Tephra deposits show a broad variety in their morphology, which represents the variable influence of eruption parameters and (post-) depositional related processes. The typical thickness of tephra layers is in the scale of centimeters but

also include several m-thick to sub-cm thin (crypto-)tephra deposits, whose lateral extent ranges from decimeter to hectometer. Visible volcanic ash layers are most commonly of bright color. Sedimentological texture properties, such as the grain-size, are in the range of ash for most tephra deposits, due to their distance to potential source volcanoes. Internal sedimentological structure of these tephra deposits, like grading, are only rarely observed due to their generally well-sorted grain-size distribution and are, if observed, commonly related to post-depositional weak water current transport processes. The

mineralogical composition of tephra layers is dominated by glass, but also primary (biotite, feldspar, zircon) and secondary



minerals (gypsum, halite) were observed. The level of details of the available morphological description of individual layers varies, as for some samples morphological and mineral compositional details are incomplete.

### 4.2  Chronology of samples

Biotites of sample PAG-T4 were $^{40}$Ar/$^{39}$Ar dated in two analytical sessions (19T04: n=10; 1-3 grains/hole; 19T11: n=20; 10 grains/hole) yielding individual weighted mean ages of 318.8 ± 18.9 ka (n=7 ages) and 298.4 ± 7.4 ka (n=18 ages), with a combined weighted mean age of 301.3 ± 13.8 ka. The weighted mean age is obtained by including as many individual analyses with mean squared weighted deviation (MSWD) < T-test statistic at 95 % confidence level. The $^{40}$Ar/$^{36}$Ar inverse isochron intercept is 297.9 ± 1.0 and overlaps with air values of Lee et al. (2006). The analytical errors in the first series are larger due 355 to small beam intensities. The radiogenic $^{40}$Ar yield is on average ~4 %. Biotites of sample IRU-1a were analyzed during the same analytical sessions (19T04: n=10; 1-3 grains/hole; 19T11: n=20; 4-5 grains/hole) as samples of PAG-T4 and their weighted mean age computation followed the same criteria. The first series yielded a weighted mean age of 188.7 ± 12.1 ka (n=6 ages) and the second series 171.2 ± 12.6 ka (n=7 ages), which gives a combined weighted mean of 180.5 ± 8.7 ka. The $^{40}$Ar/$^{36}$Ar inverse isochron intercept is 298.1 ± 0.6 and overlaps with air values of Lee et al. (2006). The radiogenic $^{40}$Ar yield 360 is very low with ~2 % on average. In the second series, parts of the experiments were discarded because the $^{40}$Ar signal was too high for the detector and runs were aborted.

In combination with the data available in literature, 42 direct tephra ages obtained by absolute dating methods and 8 indirect tephra ages obtained by luminescence or U-Th dating of the host sediment, are available for 47 of 106 samples (Table 1, 3 samples have multiple ages). Of the 42 direct ages, the majority derives from biotite $^{40}$Ar/$^{39}$Ar dating (n=29), whereas the other 365 ages were obtained by biotite K-Ar (n=10), feldspar K-Ar (n=1), feldspar $^{40}$Ar/$^{39}$Ar (n=1) and zircon U-Pb (n=1) dating methods.

The samples so far included in TephAta cover an age range from 22.90 to 0.06 Ma. Most samples (n=45) have an age < 1 Ma, but few samples (n=5) have Early Pleistocene to Miocene ages (Ritter et al., 2018; Medialdea et al., 2020). The ages of tephra layers >1 Ma suggest several distinct eruptive events, whereof only the ages of two samples partly overlap due to their 370 uncertainties of up to 21 % (CH22-NL-T1 and -T2, cf. Table 1). The $^{40}$Ar/$^{39}$Ar sanidine of tephra layer SALAR T6 age (1.297±0.018 Ma) is contradicted by a much younger IRSL host sediment age (<0.1 Ma), which is discussed in detail below. Among the <1 Ma old tephra layers, two main periods of explosive volcanic activity can be tentatively identified within the dataset (Fig. 2a). The older period includes tephra with ages of 0.9-0.5 Ma (n=7) and clustering between 0.75 and 0.60 Ma. The younger period includes samples with continuously overlapping ages between 0.43 and 0.06 Ma (n=36). One tephra 375 deposit (PAG-T4) could not be assigned to one of these two periods, as different dating techniques provide very different ages with the new $^{40}$Ar/$^{39}$Ar biotite age of 0.301±0.007 Ma and an U-Pb zircon age of 0.98 ± 0.04 Ma (Ritter et al., 2018). In general the active periods were also identified based on a partly overlapping dataset of chronological data focusing on tephra layers of the Coastal Cordillera of 20-21° S (Sepúlveda et al., 2013; Quezada et al., 2018).

Using solely the chronological data within the TephATA dataset to disentangle and identify individual volcanic events <1 Ma 380 within the two periods of volcanic activity is challenging. The heterogeneous dating techniques applied within several decades (1978-2023) used a variety of instruments, protocols, internal standards, and respective reference values, which limit the accurate differentiation of similar ages. A homogenization, e.g., through recalculation of ages based on the same reference values for mineral standards and decay constants, to increase the comparability of the complete dataset is, however, hampered by scarcity of metadata and is further limited by the fact that different dating methods have been applied. Besides the 385 uncertainties related to the different methodological approaches, also the sample-specific aspects (quantity and quality of minerals, type of target minerals, plateau vs isochrone ages) infer further ambiguity in comparing their ages. 20 samples have relatively large age uncertainties of more than >15 %, which is especially evident within the group of K-Ar ages. Furthermore, due to uncertainties exceeding the respective dated ages, some (n=2) ages have to be classified as unreliable.



The ages of 7 samples assigned to the older period (0.5-0.9 Ma) overlap among each other within their uncertainties and do
not allow a further differentiation. Among the younger sample group (<0.5 Ma), relatively small uncertainties of a few ages
(SMS-15d: 375±17 ka, GSQ-08d: 300±20 ka, GSQ-027d: 255±17 ka, TdSU: 191±5 ka and IRU-1a: 180.5±8.7 ka) indicate a
potential chronological differentiation into multiple eruptive events. All other ages of tephra layers, however, widely overlap
with the aforementioned more precise ages and among each other, which hampers the assignment of tephra to a specific
eruptive event. The age differences between some samples are within a range of 1-2 ka (e.g., GSQ-158d: 361-319 ka vs GSQ-
08d 320-280 ka; Fig. 2a), which is too small for a clear differentiation between eruptive events. Moreover, also the accuracy
of the (more precise) ages is difficult to assess as their ages are based on a single, multi-grain incremental heating experiment,
from which a plateau age was calculated. This limits the detection of sources for potential biases such as xenocryst
contamination, incorporation of altered minerals or minerals with a complex crystallization history (Hora et al., 2010; Kern et
al., 2016). Such biases are suggested for some samples, whose $^{40}Ar/^{36}Ar$ values are above the uncertainties of the atmospheric
air value and thus indicate excess Ar, so that only inverse isochrone ages could be computed (e.g., IA-11, SMS-15d).
Considering the few available total fusion ages within the dataset (PAG-T4, IRU-1a), their individual ages show a natural
scatter in ages and thus provide additional information to extrapolate more robust eruption ages, while potential biased ages
(e.g. several age populations caused by xenocrysts) can be filtered. Among the here listed samples, only very few samples
have been dated by different dating approaches to further test for potential biases. The available results indicate inconsistencies
such as by the zircon U-Pb age for tephra PAG-T4 of 0.98±0.04 Ma (Ritter et al., 2018), which is in conflict with the $^{40}Ar/^{39}Ar$
biotite age of 0.301±0.06 Ma obtained from the same sample. The youngest tephra ages included in the dataset derive from a
series of indirect infrared stimulated luminescence (IRSL) ages of the respective outcrop. The stratigraphic oldest tephra layer
of this deposit (SALAR T6) has been also dated by sanidine $^{40}Ar/^{39}Ar$ (Medialdea et al., 2020) yielding a weighted mean age
of 1.297±0. 018 Ma is based on a robust population of 25 individual total fusion ages. However, this age is much older than
the proposed IRSL maximum age <0.098±0.015 Ma and was explained by the authors as a potential reworking of the tephra
layer (Medialdea et al., 2020). New tephrostratigraphic conclusions based on new geochemical signatures (discussed below),
however, question this hypothesis. Further, the geochemical fingerprint of an overlying tephra (SALAR-T2) in the succession
likely corresponds to a tephra dated at 0.75±0.06 Ma, thus also indicating an older age of the sequence. A similar situation is
observed for a tephra found in drill cores from Salar de Uyuni, where the directly obtained biotite $^{40}Ar/^{39}Ar$ age of the tephra
is in conflict with a younger age derived from the U-series and $^{14}C$-based age model of the succession (Fornari et al., 2001;
Fritz et al., 2004). Overall, the uncertainties and inconsistencies between the different dating approaches make a chronological
differentiation between all available ages only of a tentative nature and suggest the need for additional criteria (e.g., other
dating methods, geochemical fingerprinting) for a robust separation into different eruptive events.



Figure 2: (a) Compilation of available chronological information for the samples of the Pleistocene being currently available within TephATA. For the references of the respective ages see Table 1. (b) Result of combined chronological and geochemical information illustrating the ages available for compositional groups CG 1, 2, 3, 9, and 12. The legend illustrates the different dating techniques used for the respective samples in a and b.





### 4.3 Geochemistry

For 79 of the 91 sites being listed within TephAta, glass geochemical compositions of tephra layers are available, which were obtained from 83 of the 106 samples. 65 of these samples were geochemically characterized for the first time and 16 samples had been characterized within the scope of previous projects within the CRC (May et al., 2020; Medialdea et al., 2020; Ritter et al., 2022). In addition, for two samples EPMA-WDS legacy data (2/3/5/2, AD-01; Placzek et al., 2009; Breitkreuz et al., 2014) and new measurements are available. For samples listed with an incomplete or missing geochemical characterization, the pending major, minor, and/or trace element analyses data will be complemented within the successive analytical work of the CRC1211. Comparing the legacy data with the respective new EPMA-WDS analyses reveals that the major and minor glass compositions of AD-01 are undistinguishably overlapping. However, the legacy data of sample 2/3/5/2 has significantly higher $SiO_2$ values (79.25-81.22 wt.%) compared with its reanalysis data (74.40-77.93 wt.% $SiO_2$), which results also in differences observed for the other element concentrations. A significant alkali loss and a $SiO_2$-enrichment were already noted in Breitkreuz et al. (2014), but the reason for such could not be explained. With regard to the high $SiO_2$ concentrations of the 2/3/5/2 legacy data, similarly high values were not observed for other samples within TephAta. Since no secondary reference data are given in the original article, a technical issue cannot be excluded.

Major and minor element compositions of the samples allow a general classification of their composition based on the Total Alkali vs. Silica diagram (TAS, Le Bas et al., 1986). Except for sample SALAR-T3 (basaltic-trachyandesite), all samples are dominated by glass shards having rhyolitic compositions (Fig. 3). Seven of these samples (SGL3, IRRU13, TSdU, SALAR T5, PAG17 ID4, ID14, ID24) have a more heterogeneous composition including also less evolved trachyandesitic, trachytic and/or dacitic shards. The samples of rhyolitic compositions exhibit silica concentrations which suggests a subdivision into three silica-types: type I (n=66) includes samples dominated by shards with silica concentrations between ~76-79 $SiO_2$ wt.%, type II (n=7) between ~ 73-76 $SiO_2$ wt.% and type III (n=9) represents a mix of both types having less homogenous rhyolitic compositions. This classification is also seen within the other major and minor element data of the samples investigated, as samples of the same silica type have very similar, partly overlapping major and minor oxide compositions (Fig. 3).

The majority of the rhyolitic type I samples have a similar major and minor element geochemical composition and cannot be further separated into different geochemical clusters. Some samples of type I, however, show specific variations in $TiO_2$, $FeO_{(TOT)}$, CaO, $K_2O$ and/or $Na_2O$ concentrations, which suggest a division into several clusters (Fig. 3b, d, g-i). Tephra samples of type II and type III cannot be further separated into compositional clusters based on their major element glass geochemistry, except for individual differences observed within the CaO content (Fig. 3d,e; e.g., SALAR T4/SG17/002, TJ09-PdT CH18-T3, CH18-T5).



**Figure 3: Geochemical classification and differentiation based on major and minor element glass compositions of the investigated tephra layers. (a) TAS-classification of all samples according to Le Bas et al. (1986). (b-c) TAS-diagram extraction for differentiation between type I, II and III rhyolites according to their silica content. (d, g-i) x-y oxide plots indicating a differentiation of type I rhyolites according to differences in the CaO, TiO₂, FeOtotal compositions (wt.%) and Na₂O/K₂O alkali-ratio. (d,e) x-y oxide plots indicating internal differentiation of type II and III rhyolites according to differences in the CaO (wt.%) composition of respective samples.**



A more detailed geochemical fingerprint of an eruptive event can be provided by their trace element glass shard composition (Tomlinson et al., 2012b; Pearce, 2014; Hopkins et al., 2021). Due to the more incompatible behavior of trace elements compared with major elements during magmatic differentiation (e.g. enrichment/depletion by partitioning into mineral phases
or melt during crystal fractionation/partial melting), these concentrations can provide a more diagnostic feature to distinguish between geochemical similar eruptions. For better differentiation among the identified geochemical clusters and to identify potential additional compositional differences, trace element data of 64 samples were considered. The individual samples show a mostly homogenous composition or indicate geochemical trends supporting that samples represent primary deposits and no mixture of different eruptions. Trace-element concentrations normalized to values of the primitive mantle (pyrolite;
Mcdonough and Sun, 1995) reveal for all samples (Fig. 4a-c) a relative enrichment of large-ion lithophile elements (LILE: Rb, Ba, Pb) against high field strength elements (HFSE: Ta, Nb, Hf, Zr), and among the latter an enrichment of light REE (LREE) compared to heavy REE (HREE), which is a pattern typical for the geodynamic continental arc setting. Combining the major, minor and trace element data enables the definition of 16 clusters, which are shortly discussed below and are listed in Table 2. Among the 57 individual trace-element data sets of type I rhyolites, a differentiation into 11 different clusters is suggested due
to distinct differences in their trace-element compositions (Fig. 4). Among them, 40 samples have an overlapping composition and are grouped in compositional group (CG) 1, which can be further subdivided (Fig. 4d,e) into a narrow homogenous population of 33 samples (CG1.1) and a group of seven samples generally having a similar composition, but with a wider range (CG1.2). CG 2 comprises four samples of which three are compositionally identical (CG2.1), whereas one sample shows a generally similar pattern, although with differences in composition (CG2.2). All samples can be clearly distinguished from
the other CGs by their lower HFSE and REE concentrations (Fig. 4a-c). Compositional groups 3, 4, 5, 6, 7, and 8 include one sample each, which all differ among each other (Fig. 4a-c, f-i). CG3 tephra has low Th concentrations (5-10 ppm) and a certain enrichment in HFSE (Y, Nb) and HREE compared with the other type I rhyolites, which is also observed for CG4. A characteristic depletion in Sr and Ba accompanied by elevated HFSE (except for Zr, Hf) and HREE separate CG5 from the other clusters. CG6 has slightly enriched Ba concentrations as well as HFSE and LREE, if compared to CG1. CG7 and CG8
both have a wider range of composition and have slightly lower LILE, but higher HFSE and (H+L)REE concentrations. CG9 includes four different tephra samples with a common composition, having generally higher Th concentrations (25-40 ppm, compared to 15-26 ppm of CG1.1), and further differ e.g. from CG1 by higher HFSE (Y, Nb, U) and LREE (La, Ce, Pr, Nd). CG10 has, relative to other rhyolite I tephra, lower Sr and Ba, but elevated HFSE (Y, Nb, Ta, U) and HREE concentrations, similar to tephra composition of CG11.

The individual trace element compositions of type II rhyolites (n=3) overlap, such as their major-element compositions, and thus are grouped as CG12. Their trace element compositions clearly differ from those of the other rhyolites (type I+III) as seen e.g., in lower LILE (expect for Ba, Sr) and higher HREE. Trace-element compositions of type III rhyolites (n=4) are heterogeneous among each other and allow a differentiation into four individual clusters (CG13, 14, 15, 16). CG13 has low HFSE and REE concentrations relative to other type III rhyolites. CG 14 and 15 are slightly similar and partially overlap with
CG12. However, they clearly differ in lower HFSE (Zr, Hf), higher U concentrations, lower HFSE (Zr, U), and higher Ba concentrations.



**Table 2: Overview of samples and their classification based on the Total Alkali vs. Silica composition (TAS; Le Bas et al., 1986), their assignment into rhyolites types I-III, and trace element compositional groups (TE-CG). If no trace element compositions were available, samples are listed at the end of each respective rhyolitic type. Samples which have not been geochemically investigated are listed at the bottom of the table.**

| TAS Group | CG-TE | samples | | | | |
|---|---|---|---|---|---|---|
| basaltic-trachyandesite | *not analyzed* | Salar T3 | | | | |
| rhyolite "type I" | 1.1 | 19MEJ09 | 2/3/5/2 | AVN-002d | C2-TEP-04 | C2-TEP-05 |
| | | CH17-001 | CH18-T1 | CH18-T2 | CH18-T4 | CON-1 |
| | | ESQ-1 | GSQ-027d | GSQ-08d | GSQ-158d | GSQ-53d |
| | | HU17-014 | IA-11 | IA-120 | IA-162 | IR22-001 |
| | | PAG17.2-008 | PAG17/001 | PAG17/027 | PAG18/001 | PAG18/002 |
| | | PAG18/003 | PAG18/008 | PAG-T1 | PAG-T2 | PAG-T4 |
| | | SMS-15d | Tephra RB | TOC14/001 | | |
| | 1.2 | CHU Tephra | CHU-1 | CHU-2 | IR22-002 | IRU-1a |
| | | TA-7 | TA-8 | | | |
| | 2.1 | POR 3 | TSdU | ATA19-023 | | |
| | 2.2 | PAG17/006 | | | | |
| | 3 | GSV-132 | | | | |
| | 4 | ARC1 | | | | |
| | 5 | Tephra RL N | | | | |
| | 6 | HU18/001 | | | | |
| | 7 | QP1 | | | | |
| | 8 | QP4 | | | | |
| | 9 | 10-3-5-1 | AD-01 | CTFON-4 | SALAR T2 | |
| | 10 | CH22-NL-T2 | | | | |
| | 11 | TIB4-Teph1 | TIB4-F2 | | | |
| | *not analyzed* | AD-85 | CH22-NL-T46 | HU14-008 | HU18-008 | PAG17 ID54 |
| | | SALAR T6 | SG17/001 | SG17/003 | IRRU-13 | |
| rhyolite "type II" | 12 | IQ18-001 | IQ18-002 | IV-190 | | |
| | *not analyzed* | HU18/002 | HU18/003 | SALAR T4 | SG17/002 | |
| rhyolite "type III" | 13 | CH18-T5 | | | | |
| | 15 | SALAR T1 | | | | |
| | 14 | CH18-T3 | | | | |
| | 16 | TJ09-PdT1 | | | | |
| | *not analyzed* | PAG17 ID4 | PAG17 ID14 | PAG17 ID24 | SALAR T5 | SGL3 |
| *not analyzed* | *not analyzed* | 19MEJ10 | 23MEJ1 | AN1 | CC157 | CH22-NL-T1 |
| | | CH22-NL-T3-1 | CH22-NL-T3-2 | CMI13.3 | CR-006A | GSQ-106d |
| | | IA-05 | IA-105 | IA-137 | IA-270 | IA-95 |
| | | IA-97 | IRRU-46a | IRU-15 | IS-155 | IT-307 |
| | | IT-93 | IV-189 | L5 | | |



**Figure 4: Geochemical classification and differentiation based on trace element glass compositions of investigated tephra layers. (a-c) Spidergrams demonstrating different degrees of enrichment/depletion of REE of the different composition groups. For each tephra sample the mean is given. Samples of the same composition group share the same color and are sorted according to their assignment to rhyolites I-III. Trace-element concentrations are normalized to values of the primitive mantle (pyrolite; Mcdonough and Sun, 1995). (d,e) x-y plots for trace element ratios La/Th vs. Ba/Nb to demonstrate the clustering of the different individual samples. (e) Extract of x-y plot in (d) to demonstrate the differences between CG 1.1 and CG1.2 and to CG9. (f-i) High-low plot of selected trace element ratios illustrating the differences between the different CG1-16. For each CG, all individual sample data are plotted as a stack.**


**Tephrostratigraphic and -chronological implications**

The geochemical fingerprinting approach of tephra layers from the Atacama Desert enables the identification and characterization of distinct volcanic events, which promotes the investigation of inter-site tephra correlations and their spatial dispersal. A first assessment of the combined major, minor, and trace-element compositions of the tephra layers, allows a

definition of 16 different geochemical clusters, which likely represent at least the same number of volcanic events. Some tephra layers were found in widespread archives (e.g., CG1, CG2, CG9). The widespread dispersal of these tephra layers points to major eruptive events, which have the potential to become marker horizons within the sedimentary archives of the Atacama Desert. Even though for several tephra layers no equivalents could be identified, their heterogenous geochemical clustering highlights their potential to contribute as geochemically distinct markers within the developing stratigraphic framework.

Although many of the investigated tephra layers have already been dated, the available chronological data often exhibit large uncertainties, which hamper a clear differentiation between one or multiple volcanic events. This uncertainty can be refined by including the results of the geochemical clustering to verify or falsify the indicated age differentiation, but also to identify equivalent tephra layers of the same volcanic origin. For 7 of the 16 geochemically defined clusters chronological information is available and allows the assessment of 42 individual ages. Accordingly, at least 9 volcanic events have been identified for

which no chronological information is available at present. As samples with ages >1 Ma have not been in the main focus of this exemplary dataset, an in-depth tephrostratigraphic evaluation of these older events is not possible to date. Overall, the now available chronological and geochemical data are a valuable basis for further exploring the explosive volcanic history of the region between 22 and 1 Ma. First implications are that the geochemical fingerprinting of samples <1 Ma supports the identified bifurcation of active periods within the geochronological data and further allows to characterize individual events

within the two activity periods (0.06-0.43 Ma and 0.5-0.9 Ma). For the samples >0.5 Ma, their assignment to different geochemical clusters (CG9+12) indicates at least two different eruptive events. Although their age uncertainties overlap, the combined age limits of equivalent samples suggest two eruptive events at ca. 0.75 Ma (CG9) and ca. 0.6 Ma (CG12). For the remaining samples, the clustering of their geochemical data implies three potential volcanic events.

Samples belonging to CG1 (n=40) appear to be the most widespread cluster identified so far, but their available ages do not

allow a distinct age constraint at present. Ages of CG1 include 17 ages with an age between 0.4-0.1 Ma, but also three ages exceeding 0.5 Ma (Fig. 2b). Their geochemically indistinguishable compositions imply a single volcanic eruption, but some samples with more precise ages within this group tentatively suggest a multiple eruptions (PAG-T4, SMS-15d, GSQ-08d, GSQ-027d). Thus, it cannot be fully excluded that CG1 represents a group of repeating eruptions with identical geochemical composition that occurred in short succession. Regarding the other two identified eruptive events, no precise age can be given

at present. For the older one (CG2), two differing ages of 0.191±0.005 Ma (L5/TdSU) and 0.280±0.014 Ma (ATA19-023/CR-006A) exist, whereas for the younger (CG3) only an imprecise age of 0.140±0.057 Ma (GSV-132d) is available.

In some archives, the tephrostratigraphic approach via tephra correlation also provides an independent validation of direct dating of host sediment. A tephra layer (SALAR T2) of the SALAR GRANDE hillslope section (Medialdea et al., 2020) now could be correlated with a tephra layer dated in other archives, introducing an additional age of 0.75±.06 Ma for the

succession. This age, however, is in conflict with the much younger IRSL ages from the section (<0.1 Ma), but rather supports the direct age obtained for the lowermost tephra of the succession (SALAR T6: 1.297±0. 018 Ma; Medialdea et al., 2020). The distinct geochemical compositions of all tephra layers in the succession associated with different geochemical clusters supports



a primary deposition of those layers and rather contradicts a reworking of the dated sanidine crystals form older material (Medialdea et al., 2020). In a sediment core recovered from Salar de Uyuni, the biotite $^{40}Ar/^{39}Ar$ age of a tephra layer

(0.191±0.005 Ma; Fornari et al., 2001), is questioned by a much younger, indirect U-Th age of the host sediment (0.059 Ma; Fritz et al., 2004). This tephra layer could now be geochemically correlated with two other pyroclastic deposits being located westward of Salar de Uyuni (POR-3, ATA19-023) and of which the age of ATA19-023 (resampled CR-006A: 0.280±0.014 Ma; Gardeweg and Sellés, 2013) is supporting also an older age. However, the large difference in their ages highlights the need for refining their chronology.


**Data availability**:

Table 3 (below) lists all samples included within this manuscript and provides their IGSN number and a link to download the individual geochemical data, which is stored within TephATA (https://www.crc1211db.uni-koeln.de/tephata). In addition, the full dataset (meta and chronological data overview and geochemical data including individual major, minor and trace element

glass compositions) is also made available for download at the EarthChem repository: https://doi.org/10.60520/IEDA/114209.

| Sample | IGSN/DOI | SEM/EPMA data | LA-ICP-MS data |
|---|---|---|---|
| 10-3-5-1 | 10273/GF1211S-94 | YES | YES |
| 19MEJ09 | 10273/GF1211S-4E | YES | YES |
| 19MEJ10 | 10273/GF1211S-4F | not available | not available |
| 2/3/5/2 | 10273/GF1211S--7 | YES | YES |
| 23MEJ1 | 10273/GF1211S-4H | not available | not available |
| AD-1 | 10273/GF1211S-97 | YES | YES |
| AD-85 | 10273/GF1211S-7X | YES | not available |
| AN-1 | 10273/GF1211S-77 | not available | not available |
| ARC1 | 10273/GF1211S-44 | YES | YES |
| ATA19-023 | 10273/GF1211S-3H | YES | YES |
| AVN-002d | 10273/GF1211S-33 | YES | YES |
| C2-TEP-04 | 10273/GF1211S--Y | YES | YES |
| C2-TEP-05 | 10273/GF1211S-3- | YES | YES |
| CC-157 | 10273/GF1211S-79 | not available | not available |
| CH17-001 | 10273/GF1211S-7- | YES | YES |
| CH18-T1 | 10273/GF1211S-9M | YES | YES |
| CH18-T2 | 10273/GF1211S-9N | YES | YES |

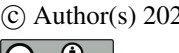



| | | | |
|---|---|---|---|
| CH18-T3 | 10273/GF1211S-9P | YES | YES |
| CH18-T4 | 10273/GF1211S-9R | YES | YES |
| CH18-T5 | 10273/GF1211S-9T | YES | YES |
| CH22-NL-T1 | 10273/GF1211S--C | not available | not available |
| CH22-NL-T2 | 10273/GF1211S--E | YES | YES |
| CH22-NL-T3-1 | 10273/GF1211S--F | not available | not available |
| CH22-NL-T3-2 | 10273/GF1211S--H | not available | not available |
| CH22-NL-T46 | 10273/GF1211S-3W | YES | not available |
| CHU Tephra | 10273/GF1211S-34 | YES | YES |
| CHU-1 | 10273/GF1211S-37 | YES | YES |
| CHU-2 | 10273/GF1211S-39 | YES | YES |
| CMI13.3 | 10273/GF1211S-7A | not available | not available |
| CON-1 | 10273/GF1211S--9 | YES | YES |
| CR-006A | 10273/GF1211S-7C | not available | not available |
| CTFON-4 | 10273/GF1211S-99 | YES | YES |
| ESQ-1 | 10273/GF1211S--A | YES | YES |
| GSQ-027d | 10273/GF1211S-3X | YES | YES |
| GSQ-08d | 10273/GF1211S--J | YES | YES |
| GSQ-106d | 10273/GF1211S-7E | not available | not available |
| GSQ-158d | 10273/GF1211S--K | YES | YES |
| GSQ-53d | 10273/GF1211S--L | YES | YES |
| GSV-132d | 10273/GF1211S-3M | YES | YES |
| HU14/008 | 10273/GF1211S-7Y | YES | not available |
| HU17-014 | 10273/GF1211S-9A | YES | YES |
| HU18/001 | 10273/GF1211S-4U | YES | YES |
| HU18/002 | 10273/GF1211S-4V | YES | not available |
| HU18/003 | 10273/GF1211S-9- | YES | not available |





| | | | |
|---|---|---|---|
| HU18/008 | 10273/GF1211S-93 | YES | not available |
| IA-05 | 10273/GF1211S-7F | not available | not available |
| IA-105 | 10273/GF1211S-7H | not available | not available |
| IA-11 | 10273/GF1211S--M | YES | YES |
| IA-120 | 10273/GF1211S--N | YES | YES |
| IA-137 | 10273/GF1211S-7J | not available | not available |
| IA-162 | 10273/GF1211S--P | YES | YES |
| IA-270 | 10273/GF1211S-7K | not available | not available |
| IA-95 | 10273/GF1211S-7L | not available | not available |
| IA-97 | 10273/GF1211S-7M | not available | not available |
| IQ18-001 | 10273/GF1211S-9E | YES | YES |
| IQ18-002 | 10273/GF1211S-9C | YES | YES |
| IR22-001 | 10273/GF1211S--T | YES | YES |
| IR22-002 | 10273/GF1211S-3A | YES | YES |
| IRRU-13 | 10273/GF1211S-4A | YES | not available |
| IRRU-46a | 10273/GF1211S-7N | not available | not available |
| IRU-15 | 10273/GF1211S-7P | not available | not available |
| IRU-1a | 10273/GF1211S-4C | YES | YES |
| IS-155 | 10273/GF1211S-7R | not available | not available |
| IT-307 | 10273/GF1211S-7U | not available | not available |
| IT-93 | 10273/GF1211S-7T | not available | not available |
| IV-189 | 10273/GF1211S-7V | not available | not available |
| IV-190 | 10273/GF1211S-7W | YES | YES |
| L5 | 10273/GF1211S-3K | not available | not available |
| PAG17 ID14 | 10273/GF1211S-4R | YES | not available |
| PAG17 ID24 | 10273/GF1211S-4T | YES | not available |
| PAG17 ID4 | 10273/GF1211S-4P | YES | not available |



| | | | |
|---|---|---|---|
| PAG17 ID54 | 10273/GF1211S--3 | YES | not available |
| PAG17.2-008 | 10273/GF1211S-4J | YES | YES |
| PAG17/001 | 10273/GF1211S--U | YES | YES |
| PAG17/006 | 10273/GF1211S-9F | YES | YES |
| PAG17/027 | 10273/GF1211S-9H | YES | YES |
| PAG18/001 | 10273/GF1211S--V | YES | YES |
| PAG18/002 | 10273/GF1211S-47 | YES | YES |
| PAG18/003 | 10273/GF1211S-49 | YES | YES |
| PAG18/008 | 10273/GF1211S-9J | YES | YES |
| PAG-T1 | 10273/GF1211S-9K | YES | YES |
| PAG-T2 | 10273/GF1211S-9L | YES | YES |
| PAG-T4 | 10273/GF1211S--4 | YES | YES |
| POR 3 | 10273/GF1211S-3J | YES | YES |
| QP1 | 10273/GF1211S-4- | YES | YES |
| QP4 | 10273/GF1211S-43 | YES | YES |
| SALAR T1 | 10273/GF1211S-3N | YES | YES |
| SALAR T2 | 10273/GF1211S-3P | YES | YES |
| SALAR T3 | 10273/GF1211S-3R | YES | not available |
| SALAR T4 | 10273/GF1211S-3T | YES | not available |
| SALAR T5 | 10273/GF1211S-3U | YES | not available |
| SALAR T6 | 10273/GF1211S-3V | YES | not available |
| SG17/001 | 10273/GF1211S-4K | YES | not available |
| SG17/002 | 10273/GF1211S-4L | YES | not available |
| SG17/003 | 10273/GF1211S-4M | YES | not available |
| SGL3 | 10273/GF1211S-3F | YES | not available |
| SMS-15d | 10273/GF1211S--W | YES | YES |
| TA-7 | 10273/GF1211S-3C | YES | YES |



| TA-8 | 10273/GF1211S-3E | YES | YES |
| Tephra RB | 10273/GF1211S--X | YES | YES |
| Tephra RL N | 10273/GF1211S-3Y | YES | YES |
| TIB4-F2 | 10273/GF1211S-4X | YES | YES |
| TIB4-TEPH1 | 10273/GF1211S-4W | YES | YES |
| TJ09-PdT1 | 10273/GF1211S-4N | YES | YES |
| TOC14/001 | 10273/GF1211S--R | YES | YES |
| TSdU | 10273/GF1211S-3L | YES | YES |

**Conclusions**

TephAta represents a new platform to support the development of a thorough tephrostratigraphic framework for the Atacama Desert in northern Chile. Its comprehensive structure is strongly aligned to intensively elaborated guidelines of the scientific tephra community (Wallace et al., 2022) and allows the storage of tephra-related information in a single online repository. It is the first online repository, which incorporates geochemical and geochronological tephra data and makes detailed outcrop and sample descriptions available, including their metadata. Besides the documentation of new samples, TephAta is also meant

to store legacy datasets, which will foster and simplify the reusability of existing datasets. Due to the automatic implementation of IGSNs for the samples, samples become citable and have a unique identification, which provides a link to incorporate datasets within other (global) databases to increase findability of data. At present the dataset includes information from 106 tephra samples. It is planned to progressively incorporate tephra data from SERNAGEOMIN and ongoing CRC-activities, which already comprise more than 400 samples. This set of samples will cover a time interval from the Early Miocene to the

Holocene and has a spatial extent from the Chilean coast to the Andes along 18-30°S.

Based on the now included first exemplary datasets within TephAta, several data-specific conclusions can be drawn. The integrity of sample and site metadata and descriptions reveals that field notes and laboratory documentation are currently recorded at different levels of detail and often lacks crucial information, probably because the samples were not collected from a tephrostratigraphic perspective. The structure given in TephAta, in combination with specific field support (sample report

templates, app-based field logs such as StraboSpot) will simplify and foster better documentation and archiving of data in the future and will facilitate public reporting of analytical details.

The geochemical characterization of the studied tephra layers reveals a mostly rhyolitic composition. Major and minor element glass composition can only be used to distinguish between some of the tephra layers. The majority of tephra layers can rather be distinguished based on their trace-element glass compositions. Based on these observed geochemical differences, at least

16 different eruptive events can be distinguished and potential regional correlations proposed. Besides these achievements, the large uncertainties of their chronological information suggest that existing ages must be carefully reviewed and supplemented, e.g., by additional ages of more confined dating methods, before used as tephrochronological tie points. Applying alternative dating methods such as zircon double-dating (Danišík et al., 2017), to exclude the influence of pre-eruptive histories of dated minerals, will allow to further assess tephra layers of similar age and geochemistry. Obtaining new chronological information

is also of interest for the numerous identified eruptive events (CG1.1, 4-8, 10, 11, 13-16), for which no chronological information is available at present. This will support the growing tephrostratigraphic framework and ensure the transfer of ages to sedimentary archives and thus help to overcome dating issues of desert sediments.



Overall, this first set of samples already illustrates the potential and usefulness of a tephrostratigraphic and tephrochronological database for the region. However, the study also indicates that the development of a regional tephrostratigraphic framework requires a substantial sample set with sufficient spatial and temporal coverage. A growing database will also contribute to a better understanding of the volcanic history of the Andes. For example, distal archives have been proven to play an important role in disentangling explosive volcanic eruptive histories (Schindlbeck et al., 2018; Leicher et al., 2023; Vineberg et al., 2024). In addition, the individual geochemical data of tephra samples provides the opportunity to study magmatic processes, such as indicated by the different degree of depletion/enrichment of characteristic elements in the presented dataset.

**Supplement:**

Supplementary Table 1:

Overview table of input fields within TephATA. For each input option within TephAta, the respective field name is given along with its overall category (e.g., "site" or "geochemistry"), the field type (e.g., text, dropdown selection, or data upload) and a short explanation of the respective field.

**Author contribution**

Conceptualization: NL, VW, BW, GB

Data curation: NL, VF

Funding acquisition, Supervision: VW, BW, GB

Resources: AQJ, PVI, FSV, GG, CB, AS, DG, LC, IRA

Software: VF, TK,

Investigation, Validation, Methodology: NL, VW, FW, ML, KK

**Writing (original draft preparation), Visualization: NL**

**Writing (review and editing): all authors**

**Competing interests**: The authors declare that they have no conflict of interest.

*Acknowledgement*

This research has been supported by the German Research Foundation (DFG) as part of Collaborative Research Centre (CRC) 1211 "Earth – Evolution at the Dry Limit", sub-project D06, A02 and Z03 (grant nos. SFB 1211/2 2020. SFB1211/3 2024). AQ, PV, FS received funding of the National Mapping Program of the SERNAGEOMIN. Gerhard Wörner, Jay Quade, Theresa Jordan, Felipe Aguilera Barraza, and Susana Layana are gratefully acknowledged for sharing sample material. Chong, G., and Jensen A. (Universidad Católica del Norte, Antofagasta) supported AS and LC during sampling. Furthermore, all (associated) CRC members who contributed sample material or assisted during field work are recognized. Reiner Kleinschrodt, Yannick Bussweiler and Hanna Cieszynski are thanked for providing access to and supporting the use of the EPMA-WDS and SEM-EDS at the University of Cologne.

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
