# Peer review of "TephAta - An online data collection of tephra data from the Atacama Desert"

_Earth System Science Data, 2025_

## Referee Comment (RC1)

**Comments on "TephAta - An online data collection of tephra data from the Atacama Desert" by N. Leicher and co-authors submitted to ESSD**

This manuscript introduces a large database of tephras from the Atacama Desert, northern Andes. The database comprises comprehensive information of the occurrence, high quality major, minor and trace element glass composition, and compilation of age determinations of about 100 individual tephra samples covering the age interval from Quaternary to Miocene. The majority of the data including Ar-Ar dating results were obtained by the authors; some literature data have been also included. The database is available in web and also as Excel files making it very convenient to use on-line or on local computer. The structure of the database is aligned with recommendations from the tephra community and thus is compatible with other databases (e.g. EarthCHEM). I want to specially emphasize that all the data is accompanied by comprehensive information on reference materials analyzed during the same analytical sessions. This is excellent example of how geochemical tephra data should be reported in any publication. The manuscript provides quite detailed analysis of the age data, compositional variations of tephras, their clustering, and possible implications for the ongoing and future research. The manuscript is very well written and without doubts is a major contribution to the tephrochronology of Andes with many important applications.

My major recommendation to authors is to try clean the database as thoroughly as possible already on this stage (when it is not too large), exclude all data of questionable quality or not supported by data on reference materials, exclude data with totals less than 90-92% and more than 101.5%, exclude obvious compositional outliers, check the correctness of web output (I specify the questions below).

Overall, I strongly recommend publication of this contribution after minor revisions, suggestions for which are placed below and mainly concern geochemical methods and results.

Line 65: Perhaps magma-crust interaction is particularly important in Andes, but in general it is not prerequisite for eruption- or volcano-specific fingerprint. Common factors resulting in volcanic glass variability is different extent of crystallization and variations in parental magma compositions.

Line 71: Are sedimentation rates high enough in the Atacama Desert to provide high temporal resolution of the archives?

Line 82: Whole-rock analysis is also technically more complicated, time-consuming and expensive compared to glass analysis by microanalytical techniques.

Line 83: not only lateral variations; crystal/glass in tephra layers are often vertically/temporally variable.

Line 165 and the above description of the database and web interface:
This is great web tool! I really like it. However, the authors may want to check the data and web output more thoroughly. For example, for randomly selected sample #TSdU, the output table contains wrong original totals (117% etc.). Supplementary table is correct.

Apparently, the 100%-normalized data in the web tables and supplementary table to this manuscript does not account for the substitution of halogens for oxygen, though it is said in the heading line of the supplementary table. Minor thing but better do it or explain clearly that Cl, F and S were not included in normalization and this is deviation from the recommended by Wallace et al. 2022 procedure.

Presumably, the authors did quality checks before entering this data into database. Or not? Was there any screen based on the values of original total? Do authors believe that totals as low as 90% and less are real, that is, due to glass hydration? One of the samples in Supplement has original total of 58% - check it.

Line 277: Please, clarify that grain-specific $SiO_2$ was applied only to samples with heterogeneous composition (as explained below). What was criteria of "heterogeneous composition"? $SiO_2$ range?

Line 296: Was Ca by LA-ICP-MS compared with Ca from EMPA data to monitor potential contamination by mineral phases? In general, how precisely was Ca content reproduced by LA-ICP-MS?

This is unfortunate that some more major/minor elements were not analyzed (Ti, Na, P etc.); this precludes rigorous analysis of the data quality and screening out of contaminated data.

After having a quick look at the data, I suggest that some variations of trace elements can be compromised by contamination during laser ablation. As an example, let us look at the composition #TIB4-TEPH1. Major elements are relatively homogeneous with $SiO_2$ range of 76.5-78% with only one obvious outlier having $Na_2O$=1.5%. Trace elements were calculated using sample-averaged $SiO_2$. The trace element and also Ca concentrations are quite variable (~2x variations, 2 points have very contrasting compositions). How the authors can prove that these variations reflect heterogeneous glass composition and do not result from contamination by mineral phases during laser ablation? Another example, analysis #52.4-PAG17.2/008-12 with high Ca and Sr, which is very likely contaminated by plagioclase during analysis.

I should mention that this is a common problem of all studies which do not analyze major elements by LA-ICP-MS. Perhaps, the authors can write a kind of disclaimer that some contaminated data may be present in the database and should be used with caution.

Line 309: One standard deviation or two? Two SD (or RSD) should be reported that corresponds to ~95% probability.

Line 340: "Crypto" refers to invisible tephra layers, disseminated in host sediments/ice.

Line 428: This is unclear why the legacy data for sample #2-3-5-2, which are not supported by standard measurements, are included in the database. I suggest to delete all questionable data from the database. This will increase the database value, consistency and reliability.

Line 453: As most of the samples are rhyolites, many trace elements behave as rather compatible elements due to strong partitioning into major and accessory mineral phases: Zr-Hf, U in zircon, Sr in plagioclase, Rb, Nb, Cs, Ba in biotite, Ba, Sr is sanidine, LREE in apatite and allanite, HREE in garnet and amphibole. Thus, the argument for using trace elements should be reformulated.

Figure 4. Lead should be placed between Ce and Pr on spider-diagrams if the elements are ordered according to their incompatibility in basaltic systems. Then, the subduction-related Pb enrichment will be seen better.

In plots 4 (f-i), what is shown by black symbols placed between 1 and 2?

Line 501: It is clear from the above that CG1, CG2 and CG9 are not "layers" but "groups of tephras of similar composition".

Line 540: Perhaps I missed some text but would it be possible to identify volcanic sources of some tephras and give this information in separate chapter?

Maxim Portnyagin

Kiel, 16-01-2026

---

## Referee Comment (RC2)

**United States Department of the Interior**

**U.S. GEOLOGICAL SURVEY**
**VOLCANO SCIENCE CENTER**

**4230 University Drive. Suite 100**
**Anchorage, Alaska 99508**

**MEMORANDUM**

**To**: Copernicus Editorial Support Team

**From**: Kristi Wallace

**Subject**: **Review comments for manuscript**: "TephAta – An online data collection of tephra data from the Atacama Desert" by Niklas Leicher et al.

Dear Editors,

Thank you for the opportunity to review this important manuscript. Below are my comments on the subject paper. I have used the review tools in Adobe Acrobat to add comments directly on the PDF (essd-2025-422_KW) and have also summarized them in bullet form below.

This manuscript describes an important regional tephra database for the Atacama Desert and Northern Andes, including detailed metadata on field sites, samples, chronology, stratigraphy, and glass geochemistry (major, minor, trace elements, and some isotopes) for 106 mid-Pleistocene tephras. I commend the authors for adopting best practices established by the global tephra community (Wallace et al., 2022) in building the TephAta database. Specifically, the authors have documented and reported key metadata (using predefined lists even), provided raw point geochemical data rather than averages, and included both primary and secondary analytical standards along with detailed methods.

This dataset represents a significant contribution to the literature and the scientific community. It will enable a wide range of applications across multiple disciplines, including volcanology (e.g., eruption histories, ash impacts, hazard assessment), archaeology (e.g., impacts, chronology), climate studies, and more. TephAta is the first accessible tephra database for this region and sets an excellent precedent for making tephra data FAIR, while encouraging future growth through contributions of new data and well-vetted legacy datasets. Furthermore, the authors have registered all stations (field sites) and samples with IGSNs, further adhering to FAIR data standards. The incorporation of tephra community best practices into the database design also establishes a strong model for future users and data collectors. Finally, the database includes a user-friendly front end, which will broaden accessibility and facilitate contributions over time.

**GENERAL COMMENTS**

- Overall, the content of the manuscript is excellent. However, I recommend a thorough copy edit to improve clarity and readability in English especially in the abstract and introduction sections (subsequent sections are in better shape). While reviewing, I found numerous sentences that could benefit from rephrasing for clarity. Addressing all of these would require considerable time, so after a few pages, I shifted my focus from detailed copy edits to broader comments and technical issues. My review therefore emphasizes the manuscript's content, overall impact, and the value of the TephAta database it introduces, rather than line-by-line language corrections.

- I appreciate the front-end user interface and the ability to plot data on maps and in geochemical space. Great job incorporating these tools, they not only enhance usability but also make it easier for me to evaluate the data within the database.

- I am not a petrologist/geochemist and therefore unable to add substantiative comments on trace element geochemistry and groupings (lines 452-495)

- **Line 26**: please include physical and chemical characterization of tephra layers (location, geochem, age, physical characteristic) and not just geochem and age especially since the term tephro'stratigraphic framework' is included in the sentence.

- **Line 30**: not sure "volcanic glass" is need in parenthesis here as the stratigraphy is documenting the tephra layer itself which contains glass, minerals, lithics and suggest indicating what you analyze (glass phase) later in the manuscript or insert "glass" geochemical composition of Pleistocene tephra layers…

- **Line 42**: remove 'so-called" and just use the term volcanic 'ash layers' or 'tephra layers', no parenthetical needed (you already did so on line 23)

- **Line 42-44**: awkward sentences and suggest revising to be more concise. Suggestion: Volcanic ash deposits are found within many types of sedimentary successions and serve as excellent chronological and stratigraphic marker horizons. This makes them highly valuable for stratigraphic and chronological studies, such as those in geoscience and archaeology (Lowe, 2011), through the application of tephrostratigraphy and tephrochronology.

- **Line 50**, these terms are becoming onerous and wonder if you should just refer to them as 'Tephrochonology' starting here? Suggest rewording: Tephochronology is a well-developed and frequently used technique for correlating and dating geological sequences and events in many regions of the world including…

- **Line 55**: please add Alaska (GeoDiva, Cameron et al.,2022). Cameron, C.E., Crass, S.W., and AVO Staff, eds, 2022, Geologic database of information on volcanoes in Alaska (GeoDIVA): Alaska Division of Geologic and Geophysical Surveys Digital Data Series 20, https://doi.org/10.14509/geodiva, https://doi.org/10.14509/30901.

- **Line 65-69**: Be clear what you mean by "pre-requisite" for tephra geochemical fingerprints as magma-crust interactions is too broad of a definition. I think you just mean that distinct geochemistry is ideal for discriminating tephras?

- **Line 274**: suggest rewording these sentences for clarity. Suggested rewording: "For homogeneous glass, $SiO_2$ concentrations obtained from EPMA-WDS data were used as the internal standard during LA-ICP-MS trace element data reduction. For samples with multiple glass populations, additional SEM-EDS analyses were performed and mapped to specific point data to provide internal standards for LA-ICP-MS trace element

calculations."

- **Line 298-300**: I am confused by this sentence. Clarify please. Also, were all major and minor oxides analyzed by EPMA or using both EPMA and SEM and if so, is this clear in the Db and have these methods been proven to be comparable on your 2 instruments?
- **Line 310**: Is this 1 standard deviation or 2?
- **Line 425**: unclear what "For samples listed with an incomplete or missing geochemical characterization…. means? Do you mean those data are planned to be collected but are not in the db now? not sure this is worth mentioning?
- **Line 426**: I assume new analyses of legacy samples are in the db and not the legacy data itself? Best to only include "best" data in the Db
- **Line 501**: do you mean that some tephras in the db were found across the Atacama Desert region? The use of "archive" made me think these tephras are found in other digital or other data archives but I think you mean they are widespread and found in the regional stratigraphy?
- **Line 527**: this is really a "tephrochronological" approach - using tephras as time stratigraphic markers by correlating undated tephras to one with a determined age.
- **Line 545**: …available for download on or after 2027-10-21. The links all go to the TephAta website and not earthchem. I appreciate that the data are uploaded to EarthChem repository as a back up in case the TephAta Db goes down and because EartthChem is a global accepted repository standard, but it would be nice to see what those data look like in EarthChem – did you use the Tephra templates provided in EarthChem?: https://earthchem.org/communities/tephra#tephra-templates. EC works well for certain data types, such as geochemistry, but it does not store ALL tephra-related information in the database, including some of the data you collected. However, the tephra templates allow you to archive this information in a spreadsheet, even if not all fields are searchable within the database.
- **Line 555**: Is IGSN automation a feature of TephAta? If so, how do you ensure legacy samples don't already have IGSN's.
- **Line 559**: unclear what this means? SERNAEOMIN has 400 samples it is ready to upload to TephAta? Can you elaborate a bit here, are they waiting to publish their data and then enter it into TephAta?
- **Line 565**: Tephra community has been working with StraboSpot for the last several year to develop a tephra module in Strabospot following the tephra community best practices for field collection and this will align well with TephAta. There will be a workshop at EGU to demo the app and to get community feedback.
- **Line 567**: be sure to include "glass" or glass phase geochemistry here to avoid confusion with whole rock composition of the parent material which of course is rarely rhyolite or at least is less evolved than the glass Geochem.
- Suggest rewording lines **567-569**: "The Geochemical characterization…."Major and minor …." **to** "Tephras in this study have predominantly rhyolitic glass compositions. Major and minor element glass compositions can differentiate only some tephra layers, whereas most layers are better distinguished by their trace-element glass compositions."
- I wonder if you are able to link any of these tephras to volcanic sources? I notice in the db itself under Tephra Equivalents that Volcano says N/A for everything.

**TECHNICAL CORRECTIONS**

- The db name, "TephATA' and 'TephAta' are used throughout and suggest changing them all to one usage. The Website looks like yet another, 'Tephata'
- **Line 36**: what is meant by Leicher N., 2027 if we are only in the year 2026)? Does this have to do with an embargo until this paper is published? The data in EarthChem don't appear to be visible.
- **Line 37**: change 'origin' to 'originating' from 91 tephra deposits…
- **Line 46**: Due to aeolian fractionation during volcanic 'ash' dispersal
- **Line 70**: change "discussed" to "thought"
- **Line 71**: delete "a" ensured good preservation of…
- **Line 73**: add or for use in extended ….
- **Line 75**: suggest rewording: "Correlation of tephra layers by their glass geochemistry is rare in the Atacama Desert region"
- **Line 80**: suggest rewording: A tephrostratigraphic framework using glass geochemistry and deposit age of widespread marker tephra layers is currently missing at the western side of the Andes.
- **Line 85**: other places this is listed as CRC1211 without a space after CRC
- **Line 93-94**: suggest rewording: Within the CRC1211, tephrochronology was used for the first time to establish a time-stratigraphich framework for the Atacama Desert.
- **Line 94**. Change, "The obtained data" to "these data are stored in a newly developed…, TephAta"
- **Line 95-97**: Suggest rewording: TephAta includes a wide variety of tephra metadata, from field context to physical and chemical characteristics.
- **Line 334**: tephra samples are source from all types (add the 's') of sediment deposits
- **Line 340**: the definition of "crypto" is invisible to the naked eye so crypto in parenthesis here is incorrect or you could update to "sub-cm to crypto tephra deposits." but sub-cm is not synonymous with crypto

**COMMENTS ON TephAta Db ITSELF (and data within)**
- I suggest that the authors focus on making sure the data in the db represents their best curated dataset and removes any questionable data. This means removing all data with low or high totals, those with missing method information and analytical standards data.
- I suggest changing the name "Geochemistry" to "Glass Geochemistry"
- It would be nice to see more **sample descriptions** in the field like that under sample TA-7 – this is currently listed under the General tab/Comment but wonder if there is a more dedicated place to include a sample description that could be text based and be a catch all for thickness, etc. Putting this in a comments field feels a little out of place and think layer description or sample description would be better.
- I am noticing that more fields show up if there is information to go in them, e.g., sample "CHU Tephra" has information under Analysis Details/Lab Split info: so it might be nice to explain in the manuscript that data only show up for fields that are filled in. Do you have templates for uploading site & sample data in addition to the

geochemical data?  I can only find the Geochem templates on the website.

- I suggest changing all instances of the name of the Db tephata to TephAta including on the website.  Left hand column of menus especially.
- **Tephra Equivalents page**:  Could this section be titled "Unique Tephra Layers" or something similar? The current wording feels a bit awkward, though it's not a major issue. Also, the wording in the green "Equivalents" box says "List of eruptions with associated tephra samples," which is helpful. Including similar explanatory text on the "Tephra Equivalents" page would improve clarity, perhaps noting that these may not represent unique or single eruptions, but rather tephras that are geochemically indistinguishable and considered equivalent or from the same source volcano at minimum.
- **Samples page:**  When performing a search on the Samples page, I filtered by $SiO_2$ range and received a table. Could you clarify what "Title" refers to? Also, table headers like MME-CM or TE-CM are not immediately clear—there seems to be space to spell these out for better understanding. Similarly, does "Site" refer to a site ID, site type, or site location? The terminology is a bit confusing, and adding clarification would improve usability. If you've already considered this extensively, I understand, but clearer labels could add value for users.  I do love that correlated tephra are in the output!  Also so many great search filters here – awesome!  Can you add IGSN to the sample search?
- **Sites page:** is title the site ID name?  might be better to call it Site ID rather than Title?  What is meant by Identifier?  I assume this is the site IGSN, can you call it that instead of identifier?
- **Clarification on IGSN Links and Registration:** When I click on the IGSN for sites or samples, it doesn't seem to lead anywhere. Could you clarify which system was used to register these site/samples? For example, I checked SESAR for sample 19MEJ09 (IGSN #10273/GF1211S-4E), but it does not appear there. I haven't tried other systems yet, but it seems there needs clarification on how samples and sites are registered with IGSN numbers.
- **Templates**: I downloaded the templates from the website rather than those uploaded to the journal. When I open them, the defined lists for data fields with drop-down menus show an arrow but no actual options. I also received the following error when opening the downloaded file: **"UNABLE TO REFRESH – We couldn't get the updated values from a linked workbook."** I am noting this in case others encounter the same issue. It might help to include a readme file with the templates or a table of definitions, so users know exactly what each field means. There are good examples and definitions for the horizontal data, but the vertical columns (e.g., *Obligation IGSN?*, *Obligation DB?*) need clarification. Also, the *Extension Keywords/drop-down lists* row shows "N/A" for all LA_ICPMS, but the spreadsheet actually contains several pick lists. Should those fields be marked as "Yes" or "No"?  The geochemical data templates are excellent!
- **Enter a new Sample:** I'd like to better understand what sample and site metadata are stored in the database. Could you make the interface for entering a new sample available for review purposes? Additionally: Is there a site and sample upload template?

**COMMENTS ON SUPPLEMENTARY MATERIALS**

    **Supplementary Table 1:**

- General comment –include the list of drop down terms for each field that has a pick list.
- General comment – this would be easier to review if it were a table formatted to show Category, Field Name, Field type, Description, Pull Down Menu all on ONE page. I had a hard time lining things up to understand the definition of each term.
- **Site** – does this mean Site type? If so can you call it "site type" and show the pick list or drop-down menus
- **Site title** – does this mean site ID? Suggest calling it "site ID" instead of "title"
- **Site Additional Comments:** I think this is where some samples have a description of the deposit…? TA-7 for example but it would be nice to have a sample description or deposit/layer description field.
- **Sample method vs sample collection method?** What is the difference? I see now that 'Sample Method' is the position of sample from within a strat layer…that is confusing and suggest replacing 'method' with 'stratigraphic position'.
- There is no sample description – seems like an oversight.
- **Morphology** – In tephra science, I typically associate morphology with grain shapes. Sup Table 1 is the first place I see what you mean, and it seems that the metadata grouped under "Morphology" actually includes two distinct types of information: the first 17 terms relate to layer characteristics and starting with "microscoped fraction," the rest are microanalytical details. To make this clearer for users, these could be divided into at least two subfields: "Layer" and "Physical Microanalytical", this breakdown would provide a more intuitive structure.
- **Stratigraphy** – these look like geochronology metadata

    **Data file supplements**

- The 2 geochemical data files are the tephra templates from EarthChem (established by the tephra community)! Great!
- **DATA_EDS_WDS:**
  - There are some metadata missing that could be included like **sample description** of the sample material analyzed which would add value; **Analyst** is listed for some but not most?; **Beam diameter** is missing for some analysis – why?; **Data line type** – what is meant by "line"-many samples have this notation? Are these the SEM-EDS data? **Sample mount** – are there not names for all mounts?
  - Some Method DOIs are links and others are not can you make them all links
  - great that the table clearly distinguishes which data come from EPMA-WDS and which from SEM-EDS. That clarification is very helpful, as it was initially unclear until I noticed the information in the far-right column (column BB).
  - I don't see where you show secondary standard analyses on the SEM-EDS? This is very important and needed to show that your SEM-EDS

and EPMA-EDS data are comparable.
- o **TephAta_Sample Info table**
  - The IGSNs all point to TephAta database and not to another IGSN repository like DataCite?

In summary, this manuscript and the TephAta database represent a valuable and well-constructed resource that will benefit multiple research communities. I recommend acceptance pending minor revisions focused primarily on improving clarity and readability and some database clean up. Thank you for the opportunity to review this important contribution.

If you have any questions regarding this review, please do not hesitate to ask for clarification.

Sincerely,
Kristi Wallace

*Kristi Wallace*

US Geological Survey/Alaska Volcano Observatory
4230 University Drive, Suite 100
Anchorage, Alaska 99508
907-786-7109
kwallace@usgs.gov